

# Biogeochemical and biological impacts of diazotroph blooms in a Low Nutrient Low Chlorophyll ecosystem: synthesis from the VAHINE mesocosm experiment (New Caledonia)

**Sophie Bonnet[1,2], Melika Baklouti[1], Audrey Gimenez[1], Hugo Berthelot[1], Ilana Berman-Frank[3]**

[1] {IRD, Aix Marseille Université, CNRS/INSU, Université de Toulon, Mediterranean Institute of Oceanography (MIO) UM 110, 13288, Marseille-Noumea, France, New Caledonia}

[2] {Institut de Recherche pour le Développement, AMU/ NRS/INSU, Université de Toulon, Mediterranean Institute of Oceanography (MIO) UM110, 98848, Noumea, New Caledonia}

 [3] {Mina and Everard Goodman Faculty of Life Sciences, Bar-Ilan University, Ramat Gan, Israel}

Correspondence to: S. Bonnet (sophie.bonnet@ird.fr)



**Abstract**
In marine ecosystems, $N_2$ fixation provides the predominant external source of nitrogen (N)
($140\pm50$ Tg N $yr^{-1}$), contributing more than atmospheric and riverine inputs to the N supply.
Yet the fate and magnitude of the newly-fixed N, or diazotroph-derived N (hereafter named
DDN) in marine ecosystems is poorly understood. Moreover, it remains unclear whether the
DDN is preferentially directly exported out of the photic zone, recycled by the microbial loop,
and/or transferred into larger organisms, subsequently enhancing indirect particle export.
These questions were investigated in the framework of the VAHINE (VAriability of vertical
and tropHIc transfer of diazotroph derived N in the south wEst Pacific) project. Triplicate
large volume (~ 50 $m^3$) mesocosms were deployed in the tropical South West Pacific coastal
ocean (New Caledonia) to maintain a stable water-mass without disturbing ambient light and
temperature conditions. The mesocosms were intentionally fertilized with ~0.8 μM dissolved
inorganic phosphorus (DIP) at the start of the experiment to stimulate diazotrophy. A total of
47 stocks, fluxes, enzymatic activities and diversity parameters were measured daily inside
and outside the mesocosms by the 40 scientists involved in the project. The experiment lasted
for 23 days and was characterized by two distinct and successive diazotroph blooms: a
dominance of diatom-diazotroph associations (DDAs) during the first half of the experiment
(days 2-14) followed by a bloom of UCYN-C during the second half of the experiment (days
15-23). These conditions provided a unique opportunity to compare the DDN transfer and
export efficiency associated with different diazotrophs. Here we summarize the major
experimental and modelling results obtained during the project and described in the VAHINE
Special issue, in particular those regarding the evolution of the main standing stocks, fluxes
and biological characteristics over the 23-days experiment, the contribution of $N_2$ fixation to
export fluxes, the DDN released to dissolved pool and its transfer to the planktonic food web
(bacteria, phytoplankton, zooplankton). We then apply our Eco3M modelling platform further
to infer the fate of DDN in the ecosystem and role of $N_2$ fixation on productivity, food web
structure and carbon export. Recommendations for future work are finally provided in the
conclusion section.



## 1 Introduction

Atmospheric dinitrogen ($N_2$) is the largest pool of nitrogen (N) on earth yet it is unavailable for most organisms that require N for growth. Biological fixation of $N_2$ (or diazotrophy) is catalyzed by the nitrogenase enzyme (encoded by the *nifH* genes) that converts the inert triple-bond $N_2$ into bioavailable ammonia ($NH_4^+$). This process has long been studied in terrestrial agriculture as it increases the yield of cultures associated with $N_2$-fixing organisms. In the ocean, diazotrophy provides the predominant external source of N ($140\pm50$ Tg N $yr^{-1}$) contributing more than atmospheric and riverine inputs (Gruber, 2004). Moreover, $N_2$ fixation acts as a potential natural fertilizer adding a source of new N that is available for non-diazotrophic primary producers and bacterioplankton especially in Low Nutrient, Low Chlorophyll (LNLC) ecosystems, where N is the proximal limiting nutrient e.g. (Moore et al., 2013). Tropical LNLC ecosystems include the vast oligotrophic subtropical gyres and represent more than 60 % of the global ocean area. $N_2$-fixing organisms (or diazotrophs) have a competitive advantage and sustain a large percentage (~50 %) of new primary production (PP) e.g. (Karl et al., 2002) in these vast ecosystems.

The non-heterocystous filamentous cyanobacterium *Trichodesmium* spp. remains the most studied marine diazotroph. Based on direct rate measurements, *Trichodesmium* accounts for a quarter to half of geochemically-derived estimates of marine $N_2$ fixation at the global scale (Mahaffey et al., 2005). Diverse cyanobacteria and bacteria also fix $N_2$ in marine waters. These include: (1) the heterocystous cyanobacteria frequently found in association with diatoms (diatom-diazotroph associations (hereafter referred to as DDAs; (Foster and O'Mullan, 2008)) efficient at exporting organic matter out of the photic zone (Karl et al., 2012), (2) unicellular cyanobacterial lineages (UCYN-A, B, and C) with a size range from 1 to 6 µm (Moisander et al., 2010), which are key oceanic diazotrophs (Luo et al., 2012) accounting for the predominant fraction of $N_2$ fixation in many tropical oceans (Bonnet et al., 2009; Montoya et al., 2004), and (3) non-cyanobacterial $N_2$-fixing bacteria and archaea that are still poorly characterized yet recent studies show they are abundant and active across the world's oceans (Farnelid et al., 2011; Farnelid and Riemann, 2008; Moisander et al., 2014).

While the role and contribution of marine $N_2$ fixation on biogeochemical cycles have been intensely investigated, a critical question that remains poorly studied is the fate of newly-fixed N, or diazotroph-derived N (hereafter named DDN) in LNLC ecosystems (Mulholland, 2007). It remains unclear whether the DDN is preferentially directly exported out of the photic zone, recycled by the microbial loop, and/or transferred into larger organisms, subsequently enhancing indirect particle export.



This question was investigated in the framework of the VAHINE (VAriability of vertical and
tropHIc transfer of diazotroph derived N in the south wEst Pacific) project. Here we
summarize the major results described in the VAHINE Special issue and integrate them to
obtain general conclusions from the experiment. In this introduction section, we first
summarize some of our knowledge regarding the fate of DDN in the ocean, describe the
ongoing technical challenges to study this question, and the specific scientific objectives of
the VAHINE project.
**1.1 Current knowledge on the fate of DDN in the ocean**
**1.1.1 DDN release to the dissolved pool**
As the biologically catalysed process of $N_2$ fixation is not entirely  efficient, diazotrophs
release some of the recently fixed $N_2$ as dissolved organic N (DON) and $NH_4^+$ to the
surrounding waters (Glibert and Bronk, 1994; Meador et al., 2007; Mulholland et al., 2006).
Several studies have reported elevated DON and $NH_4^+$ concentrations during and immediately
after *Trichodesmium* spp. blooms in the Indian (Devassy et al., 1979; Devassy et al., 1978;
Glibert and O'Neil, 1999), Pacific (Karl et al., 1992; Karl et al., 1997b), and Atlantic (Lenes et
al., 2001) oceans. Subsequent culture (Hutchins et al., 2007; Karl et al., 1992; Karl et al.,
1997a) and field studies (Benavides et al., 2013b; Konno et al., 2010; Mulholland and
Bernhardt, 2005) have quantified that diazotrophs release ~50 % of the total fixed $N_2$ to the
dissolved pool. Most of these studies were performed on the conspicuous *Trichodesmium* spp.
and were based on the difference between gross $N_2$ fixation (measured by acetylene reduction
assays) and net $N_2$ fixation (Mulholland et al., 2004) measured using the $^{15}N_2$ labelling
technique (Montoya et al., 1996). The recent modification of the $^{15}N_2$ labelling method (Mohr
et al., 2010) led to higher net $N_2$ fixation rates and potentially reduced the gap between gross
and net $N_2$ fixation. Applying the new $N_2$ fixation method and the direct measurement of the
$^{15}N$ signature on the released DON and $NH_4^+$ demonstrated low release rates from
*Trichodesmium* spp. and from three strains of UCYN-B and C (<1 % of total $N_2$ fixation)
(Berthelot et al., 2015a). Similar experiments (examining the direct $^{15}N$ measurement on
released molecules) showed low release by UCYN-C (~1 %, (Benavides et al., 2013a)).
Culture studies probably represent lower end estimates of DDN release, as in the field,
exogenous factors such as viral lysis (Hewson et al., 2004; Ohki, 1999) and sloppy feeding
(O'Neil et al., 1996) may enhance the leakage of DDN by UCYN, yet such field studies on
these organisms are rare.



### 1.1.2 Transfer of DDN to the trophic chain and impact on plankton community composition

The transfer of DDN towards the first levels of the food chain (phytoplankton, bacteria) is mainly achieved through the dissolved pool. Devassy et al. (1979) first observed that as blooms of *Trichodesmium* spp. decayed in the Indian ocean, diatom populations increased (mainly *Chaetoceros* sp.), followed by a succession of cladocerans, dinoflagellates, green algae and finally copepods. In the Atlantic, a high abundance of non-diazotrophic diatoms and dinoflagellates succeeded blooms of *Trichodesmium* spp. (Devassy et al., 1978; Furnas and Mitchell, 1996; Lenes et al., 2001), while in the pelagic waters of the Kuroshio current, *Trichodesmium* spp. and diatom abundance were positively correlated (Chen et al., 2011). These studies suggest a potential transfer of DDN from diazotrophic to non-diazotrophic phytoplankton. Actual calculations were first performed by Bronk et al. (2004), Lenes and Heil (2010) and Sipler et al. (2013), who demonstrated how the DDN released by *Trichodesmium* spp. affected the bloom dynamics of the toxic dinoflagellate *Karenia brevis* in the Gulf of Mexico. Size-fractionation of picoplankton after $^{15}N_2$ incubation also supported the idea of a DDN transfer towards non-diazotrophic plankton (Bryceson and Fay, 1981; Olendieck et al., 2007; Garcia et al., 2007), yet this method could not discriminate the DDN transfer towards non-diazotrophic picoplankton from $N_2$ fixation by picoplankton itself and thus likely overestimated the DDN transfer.

Thus, the actual transfer of DDN towards non-diazotrophic phytoplankton and bacteria remains poorly qualified and challenged due mainly to technical limitations as it requires appropriate methodologies to track the passage of DDN through the different components of microbial food web. Moreover, the planktonic groups (autotrophic *versus* heterotrophic, small *versus* large phytoplankton) that benefit the most from this DDN and develop during/after diazotroph blooms have not been identified so far despite their potential to differentially affect the structure of the trophic chain and eventually the mode of export of carbon (C) from the photic zone.

Regarding higher trophic levels, low $\delta^{15}N$ signatures measured on zooplankton indicate that DDN is transferred towards secondary producers (Montoya et al., 2002b). This transfer can be direct through the ingestion of diazotrophs (O'Neil et al., 1996; Wannicke et al., 2013a), or indirect, i.e. mediated by the dissolved N released by diazotrophs (Capone et al., 1994; Glibert and Bronk, 1994; Mulholland et al., 2004). The dissolved N (both DIN and DON) is taken up by heterotrophic and autotrophic plankton and then potentially grazed on by zooplankton, yet these pathways remain poorly explored.



The transfer of DDN to zooplankton may possibly depend on the diazotroph community
composition in the water column. Toxicity of *Trichodesmium* spp. (Kerbrat et al., 2010)
combined with poor nutritional quality (O'Neil, 1999; O'Neil and Roman, 1992) reduce
grazing pressure by copepods other than the harpacticoïd *Macrosetella gracilis*. Stable isotope
measurements performed on zooplankton suggest higher DDN uptake when the diazotroph
community is dominated by DDAs rather than *Trichodesmium* spp. (Montoya et al., 2002a).
Grazing experiments on UCYN have not been conducted so far and the potential of UCYN as
a conduit of DDN into marine food webs remains unexplored.
**1.1.3 Export of DDN out of the photic zone**
Low $\delta^{15}$N signatures in particles from sediment traps in the tropical North Pacific suggests
that at least part of the DDN is ultimately exported out of the photic zone (Karl et al., 2012;
Karl et al., 1997b; Scharek et al., 1999a; Sharek et al., 1999b). The export of DDN may either
be direct through sinking of diazotrophs, or indirect, through the transfer of DDN to non-
diazotrophic plankton in the photic zone, that is subsequently exported. While it has been
demonstrated that DDAs directly contribute to particle export (Karl et al., 2012; Subramaniam
et al., 2008; Yeung et al., 2012), the DDN export efficiency appears to depend on the
diazotroph community composition present in surface waters. The positive buoyancy of
*Trichodesmium* spp. probably prevents its downward flux and settling in sediment traps
(Capone et al., 1997; Walsby, 1992), although programmed cell death (PCD) causing bloom
demise can cause rapid export of *Trichodesmium* biomass to depth (Bar-Zeev et al., 2013;
Berman-Frank et al., 2004; Spungin et al., 2016). In the north-east Pacific, when the
diazotrophic community was dominated by UCYN-A and *Trichodesmium* spp., N$_2$ fixation
contributed ~10 % of the export (White et al., 2012); when DDAs dominated the diazotrophic
community they contributed ~44 % of export production, thereby suggesting that DDAs have
a higher export efficiency compared to *Trichodesmium* spp. and UCYN-A. Despite their
recent recognition as key oceanic diazotrophs (Luo et al., 2012), the export efficiency of
UCYN from other lineages (UCYN-B and UCYN-C) is currently undetermined as no
published studies of natural UCYN blooms and their fate in the ocean are to date available.
The determination of direct *versus* indirect export requires diazotroph quantification in both
the water column and in sediment traps in addition to clarifying the actual transfer of DDN to
the different groups of autotrophic and heterotrophic plankton. Few studies have thus focused
on the direct coupling between N$_2$ fixation and particulate export in general (see references
above). Ideally such studies require the successful encounter of an oceanic diazotroph bloom,



deployment of sediment traps, and long-term (several weeks) monitoring of the
biogeochemical characteristics of the water body influenced by the bloom, which are rarely
accomplished. The patchy distribution of diazotrophs in the surface ocean (Bombar et al.,
2015), the temporal lag between production and export, and hydrodynamic features that may
decouple production in surface and export below the photic zone (Buesseler et al., 2007) also
make these studies very challenging.
**1.2 Scientific objectives of the VAHINE project**
Thus, the main scientific objectives of the VAHINE project were:
i) To quantify the DDN which enters the planktonic food web. Is DDN preferably transferred
to large size (e.g. diatoms), small size (pico-, nanophytoplankton) phytoplankton, or to the
microbial food web? What percentage of DDN is transferred to zooplankton? Does it depend
on the diazotroph community composition?
ii) To investigate how the development of diazotrophs influences the subsequent diversity,
gene expression, and production of primary producers, heterotrophic bacterioplankton, and
subsequently the zooplankton abundance
iii) To examine whether different functional types of diazotrophs significantly modify the
stocks and fluxes of the major biogenic elements (C, N, P)?
iv) To elucidate whether the efficiency of particulate matter export depends on the
development of different functional types of diazotrophs? Is this export direct (through the
sinking of diazotrophic cells) or indirect (through the transfer of DDN to non-diazotrophic
plankton that is subsequently exported)?
To achieve these goals and concurrently determine $N_2$ fixation and particle export, we isolated
large water masses containing ambient planktonic communities by deploying three large-
volume (~50 m$^3$) mesocosms (Bonnet et al., 2016) thereby maintaining a stable water-mass
without disturbing ambient light and temperature conditions. The experimental location in the
southwestern Pacific region was chosen as in this area some of the highest rates of oceanic $N_2$
fixation occur (Bonnet et al., 2015b; Messer et al., 2015). Additionally, to enhance $N_2$
fixation, the mesocosms were intentionally fertilized with dissolved inorganic phosphorus
(DIP). The experiment lasted 23 days and was characterized by a dominance of DDAs during
the first half of the experiment (days 2-14) and a bloom of UCYN-C during the second half of
the experiment (days 15-23), providing a unique opportunity to compare the DDN transfer



and export efficiency associated with specific diazotrophs in this experimental system. Some
additional process experiments performed on *Trichodesmium* spp. which bloomed outside the
mesocosms on the last two days are also presented here.
Below, we summarize the scientific strategy used in this study, as well as some of the major
results obtained during this project and propose some scientific perspectives for the future.
**2   Scientific strategy**
**2.1 Brief description of the mesocosms and study site**
The large-volume (~50 m$^3$) mesocosm experiment was undertaken in New Caledonia, located
1500 km east of Australia in the Coral Sea (southwestern tropical Pacific, Fig. 1). Three
replicate polyethylene and vinyl acetate mesocosms (diameter 2.3 m, height 15 m, volume
~50 m$^3$, Fig. 2) were deployed 28 km off the coast of New Caledonia at the entrance to the
Noumea coral lagoon (22°29.073 S - 166°26.905 E) for 23 days from January 13$^{th}$ to February
6$^{th}$ (austral summer). The New Caledonian lagoon has been chosen as it is a well-studied
environment (Special issue Marine Pollution Bulletin 2010 (Grenz and LeBorgne, 2010))
submitted to high oceanic influence (Ouillon et al., 2010) and harbouring typical oligotrophic
conditions during the summer season (NO$_3^-$ concentrations <0.04 µmol L$^{-1}$ and chlorophyll a
(Chl *a*) ~0.10-0.15 µg L$^{-1}$ (Fichez et al., 2010). Primary productivity is N-limited throughout
the year (Torréton et al., 2010), giving diazotrophs a competitive advantage. New Caledonian
waters support high N$_2$ fixation rates (151-703 µmol N m$^{-2}$ d$^{-1}$, (Garcia et al., 2007)), as well
as high *Trichodesmium* spp. (Dupouy et al., 2000; Rodier and Le Borgne, 2010, 2008), and
UCYN abundances (Biegala and Raimbault, 2008), therefore representing an ideal location to
implement the VAHINE project and study the fate of DDN in the marine ecosystem.
DIP availability can control N$_2$ fixation in the southwestern Pacific (Moutin et al., 2008;
Moutin et al., 2005), hence the mesocosms were intentionally fertilized with ~0.8 µM DIP
(KH$_2$PO$_4$) the evening of day 4 to alleviate any potential DIP limitation and promote N$_2$
fixation and even diazotroph blooms for the purpose of the project.
The mesocosms used for this study are well suited for conducting replicated process studies
on the first levels of the pelagic food web (Bonnet et al., 2016; Guieu et al., 2010; Guieu et
al., 2014). They are equipped with sediment traps allowing the collection of sinking material.
Due to the height of the mesocosms (15 m), they do not represent processes occurring in the
full photic layer but allow studying the dynamics of C, N, P pools/fluxes and export
associated with the plankton diversity in the same water mass, and comparing these dynamics.
before/after the DIP fertilization, and under contrasted conditions regarding the diazotroph





community composition (cf below). Detailed surveys performed in LNLC environments
revealed that temperature and light conditions are not affected by the presence of the
mesocosms compared to surrounding waters (Bonnet et al., 2016; Guieu et al., 2010; Guieu et
al., 2014). These studies also revealed a good replicability of stocks, fluxes and plankton
diversity measurements among the replicate mesocosms. Hence, the discussion below will
consider the average between the three mesocosms deployed in this study.
**2.2 Sampling strategy and logistics**
A complete description of the mesocosms design and deployment strategy is given in the
introductory article (Bonnet et al., 2016). In total, over 47 stocks, fluxes, enzymatic activities
and diversity parameters were measured daily by the 40 scientists involved in the project.
Protocols for each measured parameter are detailed in the specific contributions to this special
issue and will not be described here. Modelling has also accompanied all steps of the project
(see Gimenez et al. (2016) and section 5 below).
Sampling for stocks, fluxes and plankton diversity measurements was performed daily at 7 am
in each of the three mesocosms (M1, M2 and M3) and in surrounding waters (hereafter called
'lagoon waters') from day 2 (January 15[th], the day of the mesocosms closure) to day 23
(February 6[th]) at three selected depths (1, 6 and 12 m) to study the vertical variability within
and in lagoon waters. For flux measurements, bottles were incubated on an in situ mooring
line at the appropriate sampling depth set up close to the mesocosms. Vertical CTD profiles
were then performed daily at 10 am in every mesocosm and in lagoon waters using a SBE 19
plus Seabird CTD to obtain the vertical profiles of temperature, salinity and fluorescence.
Finally, sediment traps were collected daily by SCUBA divers at 10:30 am, see details in
Bonnet et al. (2016).
**3  Evolution of the main standing stocks, fluxes and biological**
**characteristics during the VAHINE experiment**
Initial hydrological and biogeochemical conditions (i.e. conditions in ambient waters the day
of mesocosms deployment - January 13[th], day 0) were typical of those encountered in the
oligotrophic Noumea lagoon during austral summer conditions (Fichez et al., 2010; Le
Borgne et al., 2010), with seawater temperature of 25.5°C, surface salinity of 35.15, $NO_3^-$-
depleted waters ($0.04\pm0.01$ µmol L$^{-1}$), low DIP concentrations ($0.04\pm0.01$ µmol L$^{-1}$), and Chl
$a$ concentrations of 0.20 µg L$^{-1}$. $N_2$ fixation rates were $8.70\pm1.70$ nmol N L$^{-1}$ d$^{-1}$ and the
diazotroph community was dominated by DDAs (het-1 $3.1 \times 10^4$ *nifH* copies L$^{-1}$ and het-2 1.2



x10$^4$ *nifH* copies L$^{-1}$) as well as UCYN-A2 (1.5 x 10$^4$ *nifH* copies L$^{-1}$) and UCYN-A1 (5.6 x
10$^3$ *nifH* copies L$^{-1}$), which together accounted for 95 % of the total *nifH* pool in the lagoon
waters prior to the mesocosms closure (Turk-Kubo et al., 2015).
During the 23-days VAHINE mesocosm experiment, three major periods could be defined
based on the main C, N, P stocks and fluxes (Berthelot et al., 2015b) and on the identity of the
most abundant diazotrophs that developed in the mesocosms (Turk-Kubo et al., 2015): **P0**
from days 2 to 4 (i.e. prior to the DIP fertilization that occurred on the evening of day 4), **P1**
from days 5 to 14, and **P2** from days 15 to 23 (Figs. 3 and 4). Figure 3 reports the main
hydrological and biogeochemical parameters during the experiment. Figure 4 provides a
synoptic view of the main changes (positive, negative, neutral) in the major stocks, fluxes,
and plankton community composition measured during P1 and P2 respectively.
Seawater temperature (Fig. 3) gradually increased both inside and outside the mesocosms
over the 23-days of the experiment from 25.5°C to 26.2°C on day 23, which is the general
trend observed during austral summer conditions (Le Borgne et al., 2010). The water column
was well homogenized inside the mesocosms throughout the experiment (Bonnet et al., 2016).
NO$_3^-$ concentrations remained close to detection limit of conventional micromolar methods
(0.02 µmol L$^{-1}$) both inside and outside the mesocosms throughout the 23 days of the
experiment (Fig. 3). The low (0.04 µmol L$^{-1}$) DIP concentrations measured during P0
increased in the mesocosms right after the fertilization up to ~0.8 µmol L$^{-1}$, then decreased
quickly to reach values close to initial DIP concentrations (~0.04 µmol L$^{-1}$) at the end of the
experiment.
As a major objective of the experiment was to study the development of diazotroph blooms
and the fate of DDN, investigation of the biological response was focused on diazotrophs and
their subsequent influence on biological and biogeochemical signatures. N$_2$ fixation rates
tripled between P1 and P2, to reach extremely high rates during P2 (27.3±1.0 nmol N L$^{-1}$ d$^{-1}$
on average and up to 70 nmol N L$^{-1}$ d$^{-1}$ (Bonnet et al., 2015a)) (Fig. 3), ranking among the
highest rates reported in marine waters (Luo et al., 2012). The diazotroph community
composition was dominated by DDAs during P1, and a bloom of UCYN-C occurred during
P2 (Fig. 4). Standing stocks of Chl *a* and PON increased by a factor of 3 and 1.5 between P1
and P2 and subsequently, export of PON dramatically increased (by a factor of 5) in the
mesocosms during P2 (Fig. 3). These results emphasize that the experimental mesocosm setup
provided ideal conditions to study the fate of DDN associated with different diazotroph
communities (DDAs *versus* UCYN-C).



The synoptic view of the mesocosm dynamics (Fig. 4) indicates that after the DIP
fertilization, DIP concentrations and DIP turn-over time increased significantly during P1, and
alleviated P-limitation in the microbial communities as reflected in the significant decline in
alkaline phosphatase activity (APA). The major biomass-indicative standing stock parameters
(Chl $a$, POC, PON, POP) did not increase immediately after the DIP fertilization (P1) but
during P2 (see below). Only PP increased significantly by a factor of 2 during P1, associated
with a significant increase in $N_2$-fixing DDAs and *Prochlorococcus* abundances. During P1,
enhanced DIP availability enabled non-diazotrophic organisms with lower energetic
requirements and higher growth rates such as *Prochlorococcus* to outcompete the diazotrophs
in the mesocosms via utilization of recycled N derived from $N_2$ fixation (Bonnet et al., 2015a).
Thus, while PP increased, $N_2$ fixation rates decreased significantly after the DIP spike.
During P2, diazotrophy was characterized by the significant increase in UCYN-C abundances
that reached up to 7 x $10^5$ *nifH* copies $L^{-1}$, concomitant with the utilization of DIP and the
significant decline in DIP concentrations, DIP turn-over time and a parallel increase of total
APA. In all three mesocosms, the increase in UCYN-C abundances coincided with the day at
which the DIP turnover time declined below 1 d, indicative of DIP limitation (Berthelot et al.,
2015b; Moutin et al., 2005). UCYN-C may have also utilized dissolved organic phosphorus
(DOP) as a P source (Bandyopadhyay, 2011), driving the significant decline in DOP
concentrations observed during P2 ((Berthelot et al., 2015b), Fig. 4). The mesocosm approach
also enabled the calculation of *in situ* growth rates for UCYN-C, which were up to 2 $d^{-1}$
during P2, i.e. higher than growth rates of other diazotrophic phylotypes during P2 (Turk-
Kubo et al., 2015), indicating that under $NO_3^-$ depletion and low DIP availability, UCYN-C
was the most competitive diazotroph in the mesocosms.
Under the high $N_2$ fixation conditions encountered during P2 (27.3±1.0 nmol N $L^{-1}$ $d^{-1}$), all
standing stocks (Chl $a$, POC, PON, POP) increased in the mesocosms, together with PP and
BP (Fig. 4). The corresponding $NO_3^-$, DIP, DON and DOP stocks for P2 decreased, indicating
active consumption by the planktonic communities. As no external supply of $NO_3^-$ was
provided to the enclosed mesocosms, we calculated that the consumption of the $NO_3^-$ stock
initially present in the mesocosms (0.04 µmol $L^{-1}$) represented less than 11 % of the integrated
$N_2$ fixation rates. Therefore, $N_2$ fixation supplied nearly all of the new production during the
experiment. Our results demonstrate that in oligotrophic N-depleted systems, diazotrophs can
provide enough new N to sustain high PP rates (exceeding 2 µmol C $L^{-1}$ $d^{-1}$) and high biomass
(~ 10 µmol $L^{-1}$ of POC and 0.7 µg $L^{-1}$ of Chl $a$), as long as DIP does not limit $N_2$ fixation





(Berthelot et al., 2015b). Furthermore, during P2, DON provided an additional N source for
non-diazotrophic phytoplankton and bacteria (Berthelot et al., 2015).
The time lag between the DIP fertilization and the increase in biogeochemical stocks/fluxes
was 10 days, indicating that 10 days were necessary for $N_2$ fixation to sustain the high
production rates observed, and to see an effective accumulation of biomass. Our results
demonstrate the restricted applicability of nutrient-addition experiments in small-volume
microcosms (several liters) mostly limited to 24-72 h incubations that are typically employed
to assess nutrient limitations on plankton growth in the ocean, e.g. (Moore et al., 2013). If
indeed a longer time scale (weeks) is required to study nutrient limitation of plankton in
marine ecosystems, then large-volume mesocosms, such as we demonstrate here, would be
more suitable (Gimenez et al., 2016).
Concurrent with the development of diazotrophic (UCYN-C) populations, the abundance of
*Synechococcus*, pico-eukaryote and nano-eukaryote primary producers also increased at the
end of P2 (i.e. around day 16) (Leblanc et al., 2016). The non-diazotrophic diatoms responded
rapidly (i.e. around day 10-11) and increased to bloom values (100 000 cells $L^{-1}$)
simultaneously with the UCYN-C bloom on days 15-16 and prior to the increases in the pico-
and nanophytoplankton (Pfreundt et al., 2015; Van Wambeke et al., 2015). This increase was
paralleled by a drastic change in the diatom community structure, which became almost
monospecifically dominated by *Cylindrotheca closterium*. Despite the significant increase in
BP during P2 and enrichments in the 16S transcripts of specific bacterial groups (Pfreundt et
al., Submitted), the total abundance of heterotrophic bacteria did not (Van Wambeke et al.,
2015), probably due to grazing. Finally, no consistent temporal pattern in zooplankton
biomass was detected over the course of the experiment (Hunt et al., 2016), although changes
were observed regarding the contribution of DDN to zooplankton biomass (see below).
**4. Tracking the fate of $N_2$ fixation**
**4.1. Contribution of $N_2$ fixation to export fluxes**
We specifically utilized the mesocosm approach to answer whether the composition of the
diazotroph community influenced the subsequent export of particulate matter and how.
During P1, DDAs dominated the diazotroph community. For this time period, the biomass
indices (Chl *a*, POC, PON, POP) were stable within the mesocosms (Fig. 3, 4), suggesting
that the DDN associated with DDAs remained within the symbiotic associations (i.e. was
poorly transferred to the rest of the planktonic community). Moreover, the amount of recently
fixed $N_2$ equaled that of exported PON, suggesting that the recently fixed $N_2$ by DDAs was



rapidly exported (Fig. 5a) as also observed for DDAs in the tropical North Pacific at Station
ALOHA (Karl et al., 2012). DDAs such as het-1 (*Richelia* in association with the diatom
*Rhizosolenia* spp.), which dominated the DDA community during P1 in the mesocosms
(Turk-Kubo et al., 2015) have indeed been shown to sink at high rates in the ocean (Scharek
et al., 1999a).
During P2 and the UCYN-C bloom, the increases in Chl *a*, POC, PON, and POP
concentrations in the mesocosms suggest that a fraction of the recently produced biomass
sustained by $N_2$ fixation remained in the water column. The mesocosms enabled us to
determine whether export associated with diazotrophs was direct (through the sinking of
diazotrophic cells) or indirect (through the transfer of DDN to non-diazotrophic plankton that
is subsequently exported). The direct export of UCYN has rarely been studied (White et al.,
2012). Yet, UCYN contribution to vertical flux and export was assumed to be lower than the
contribution of DDAs due to their small size of (1 to 6 µm) and low sinking rates compared to
DDAs (up to 500 µm comprised of dense silica shells). qPCR quantification of diazotrophs in
the sediment traps  revealed that ~10 % of UCYN-C from the water column was exported to
the traps daily, representing as much as 22.4±5.5 % of the total POC exported at the height of
the UCYN-C bloom (Bonnet et al., 2015a). Mechanistically, the vertical downward flux was
enabled by the aggregation of the small (5.7±0.8 µm) UCYN-C cells into large (100-500 µm)
aggregates, the size of which increased with depth (Fib. 5b) possibly due to a sticky matrix
composed also of transparent exopolymeric particles (TEP), which concentrations increased
during P2 (Fig. 4) (Berman-Frank et al., 2016). These data, reported for the first time from the
VAHINE experiment (Bonnet et al., 2015a), emphasize that despite their small size relative to
DDAs, UCYN-C are able to directly export organic matter to depth, indicating that these
small organisms should be considered in future biogeochemical studies.
The direct export of UCYN-C and other diazotrophs could not solely explain the very high
exported matter observed during P2 (Bonnet et al., 2015a), suggesting another way of export
during that period. An experiment performed during the UCYNC bloom using nanoSIMS
demonstrated that a significant fraction of DDN (21±4 %) was quickly (within 24 h)
transferred to non-diazotrophic plankton (Bonnet et al., 2015a), revealing that $N_2$ fixation was
fuelling non-diazotrophic plankton growth in the water column (Fig. 5b), suggesting an
indirect export pathway in addition to the direct export of UCYN-C. The fact that UCYN-C
fuelled non-diazotrophic plankton during P2 is consistent with the increase in biomass
indicators as well as the increase in non-diazotrophic phytoplankton abundances (diatom and
picoplankton) simultaneously with or after the UCYN-C bloom during P2.





The high export efficiency associated with the UCYN-C bloom compared to the one
associated with the DDAs during VAHINE was also indicated by *e*-ratio calculations, which
quantify the efficiency of a system to export POC relative to PP. During P2, the *e*-ratio was
significantly ($p<0.05$) higher (i.e., during the UCYN-C bloom; $39.7\pm24.9$ %) than during P1
(i.e., when DDAs dominated the diazotrophic community; $23.9\pm20.2$ %) (Berthelot et al.,
2015b). $\delta^{15}N$ measurements on DON, PON and particles from sediment traps further
substantiated these results with a significantly ($p<0.05$) higher contribution of $N_2$ fixation to
export production during P2 ($56\pm24$ % and up to 80 % at the end of the experiment) compared
to P1 ($47\pm6$ %) (Knapp et al., 2015). The contribution of $N_2$ fixation to export (up to 80 %)
was very high in our study compared with reports from other tropical and subtropical regions
where active $N_2$ fixation contribute 10 to 25 % to export production (e.g. (Altabet, 1988;
Knapp et al., 2005)). This is consistent with the extremely high $N_2$ fixation rates measured in
the mesocosms (up to 70 nmol N $L^{-1}$ $d^{-1}$) compared to those measured in other regions (Luo et
al., 2012).
The export associated with *Trichodesmium* spp. has not been studied in the present mesocosm
experiment as only limited numbers of *Trichodesmium* spp. were counted in the mesocosms.
Its potential for export is discussed below based on parallel studies from the region and
intensive short-term experiments on surface blooms of *Trichodesmium* that appeared outside
the mesocosms on days 22-23 (Spungin et al., 2016).
**4.2. DDN release and transfer to the food web**
**4.2.1 DDN release and transfer to non-diazotrophic phytoplankton and bacteria**
As part of VAHINE, we assessed  the quantity of DDN entering the planktonic food web as a
function of the dominant diazotroph players, and examined which planktonic communities
benefited the most from the DDN (i.e. small *versus* large phytoplankton, microbial food
web?).
Diazotrophs transfer DDN to phytoplankton and heterotrophic prokaryotes via the dissolved
N pool (DON and $NH_4^+$). During the maximal abundance of UCYN-C, UCYN-C were
responsible for $90\pm29$ % of total $N_2$ fixation rates in the mesocosms (Bonnet et al., 2015a) and
the DDN released to the dissolved pool (based on the direct measurement of the isotopic
signature ($^{15}N$) of the total dissolved N according to the denitrifying method (Knapp et al.,
2005)) accounted for $7.1\pm1.2$ to $20.6\pm8.1$ % of gross $N_2$ fixation (Bonnet et al., 2015a). This
proportion is higher than that reported for UCYN-C in monospecific cultures using an
equivalent method ($1.0\pm0.3$ to $1.3\pm0.2$ % of gross $N_2$ fixation (Benavides et al., 2013a;



Berthelot et al., 2015a). In the natural waters of the mesocosms, a diverse diazotroph
community was found at the same time as UCYN-C (Turk-Kubo et al., 2015), and probably
contributed to some DDN release. Additionally, exogenous factors such as viral lysis
(Fuhrman, 1999) and sloppy feeding (O'Neil and Roman, 1992) occur in natural populations
and could enhance N release compared to the mono-culture studies. To our knowledge, these
data are the first reported of DDN released in a UCYN bloom.
The physiological state of cells probably plays a critical role in the quantity and availability of
DDN to the microbial communities as demonstrated in a study (applying identical
methodology) from two naturally-occurring blooms of *Trichodesmium* spp. in the same area
(New Caledonian lagoon) (Bonnet et al., Accepted). DDN release from these blooms was
slightly higher (bloom 1: 20±5 to 48±5 % and bloom 2: 13±2 to 28±6 % of gross $N_2$ fixation)
compared to UCYN-C (Bonnet et al., Accepted). *Trichodesmium* spp. bloom 1 was decaying,
leading to high DDN release rates and high $NH_4^+$ accumulation (up to 3.4 μM) in the
dissolved pool, which was not observed during bloom 2 when *Trichodesmium* spp. were in
exponential growing phase. The importance of physiological status rather than specific
diazotroph types was further substantiated in culture study showing no significant differences
in DDN release between *Trichodesmium* spp. and three strains of UCYN-B and C (Berthelot
et al., 2015a)
Previous comparisons between gross and net $N_2$ fixation rates indicated high DDN release
rates for oceanic populations of *Trichodesmium* spp. (40-50 % of gross $N_2$ fixation on
average, and up to 97 %, (Mulholland, 2007) and references therein). The physiological status
of these populations may have influenced the fluxes. Furthermore, the values could reflect a
methodological overestimation due to the use of the $^{15}N_2$ bubble method (Montoya et al.,
1996) that may lead to greater differences between gross and net $N_2$ fixation (see
introduction). Currently, direct measurement of the $^{15}N$ signature of the dissolved N pool
itself (either the TDN pool through the Knapp et al. (2005) method or both the $NH_4^+$ and the
DON using the Slawyk and Raimbault (1995) method) appears the preferred method to
accurately quantify the amount of DDN released by diazotrophs in the dissolved pool
(Berthelot et al., 2015a).
Once released in the form of $NH_4^+$ and/or DON, DDN can be taken up by surrounding
planktonic communities. Experimental evidence from nanoSIMS experiments during
VAHINE indicate that 21±4 % of the $^{15}N_2$ fixed during the UCYN-C bloom was transferred
to the non-diazotrophic plankton after 24 h of incubation (Bonnet et al., 2015a). Among these





21±4 %, 18±3 % was transferred to picoplankton (including both pico-phytoplankton and
heterotrophic prokaryotes) and 3 % to diatoms (Fig. 5b), suggesting that picoplankton would
be more competitive than diatoms using DDN, which is consistent with the increase in
*Synechococcus* and pico-eukaryote abundances by a factor of two following the UCYN-C
bloom (Leblanc et al., 2016; Pfreundt et al., 2015). The short-term nanoSIMS experiment was
performed on day 17, when pico- and nanoplankton dominated the phytoplankonic biomass
and diatom abundances declined probably due to DIP limitation (Leblanc et al., 2016).
Picoplankton can efficiently utilize low DIP concentrations (Moutin et al., 2002) and/or can
use alternative DOP sources (Pfreundt et al., Submitted; Van Wambeke et al., 2015), which
may explain why they were the first beneficiaries of the DDN from UCYN-C at that time of
the mesocosm experiment, although we cannot exclude that diatoms had also benefited from
the DDN from UCYN-C but earlier in the experiment (between days 10-11 and days 15-16
when they reached bloom values of ~100 000 cells $L^{-1}$), when the DIP turn-over time was still
higher than 1 d (indicative of no DIP limitation, (Berthelot et al., 2015b)).
A significant increase of both PP and BP during P2 (Fig. 2) suggests that both autotrophic and
heterotrophic communities benefited from the DDN (Bonnet et al., 2015a). Calculations based
on C:N molar ratios show that $N_2$ fixation may have provided ~30 % of the N demand of the
N-limited bacteria during P2 (compared to ~20 % during P1), the rest being likely provided
by detritus and DON (Van Wambeke et al., 2015), which concentrations decreased during the
23 days (Berthelot et al., 2015b). The biological system inside the mesocosms was net
autotrophic during VAHINE, with an upper error limit close to the metabolic balance between
autotrophy and heterotrophy (Van Wambeke et al., 2015). The weak (during P2) or absent
(during P1) correlations between BP and $N_2$ fixation rates and the tightly coupled
relationships between BP and Chl *a* concentrations, and between BP and PP suggests that $N_2$
fixation stimulated autotrophic communities and these subsequently stimulated heterotrophic
prokaryotes through the production and release of dissolved organic matter including C
(DOC) (Van Wambeke et al., 2015).
In a recent study performed at the VAHINE study site, (Berthelot et al., 2016) compared the
DDN transfer efficiency to several groups of non-diazotrophic plankton as a function of the
diazotroph groups dominating the community (*Trichodesmium* spp. *versus* UCYN-B *versus*
UCYN-C). Simulated blooms of *Trichodesmium* spp., UCYN-B and UCYN-C grown in
culture added to ambient lagoon communities reveal that the primary route of transfer of
DDN towards non-diazotrophs is $NH_4^+$, and DON mainly accumulates in the dissolved pool,
whatever the diazotroph considered. In all cases, the presence of diazotrophs stimulated




biomass production of non-diazotrophs, with heterotrophic prokaryotes the main beneficiaries
of the DDN followed by diatoms and picophytoplankton. NanoSIMS analyses revealed that
heterotrophic prokaryotes were highly $^{15}$N-enriched, confirming they can directly benefit from
the DDN (Berthelot et al., 2016). Further studies are needed to study the indirect stimulation
of heterotrophic prokaryotes through the release of DOC by diazotrophs and non-diazotrophic
phytoplankton that has been stimulated by the DDN.
Similar experiments ($^{15}$N$_2$ labelling, flow cytometry cell sorting and nanoSIMS) performed on
three naturally-occurring *Trichodesmium* spp. blooms in the southwestern Pacific illustrated
that DDN was predominantly transferred to diatoms whose abundance increased from 1.5 to
15-fold during and after the *Trichodesmium* spp. blooms (Bonnet et al., Accepted). The results
from these small-scale experiments indicate that under realistic conditions the extensive
oceanic blooms of *Trichodesmium* spp. (reaching tens to thousands of km$^2$), the high amounts
of DDN can fuel successively large diatom or dinoflagellate blooms (Bonnet et al., Accepted;
Devassy et al., 1979; Lenes et al., 2001), whose efficient export rates (Nelson et al., 1995) can
contribute to a large indirect downward flux of organic matter (Fig. 5c).
Direct export flux of *Trichodesmium* spp. blooms may also occur in cases where rapid (< 2 d)
bloom mortality occurs via a programmed cell death (PCD) process that is induced under
environmental stressors (e.g. Fe limitation, oxidative stress) or physiological status (stationary
phase) (Berman-Frank et al., 2004; Berman-Frank et al., 2007). PCD in *Trichodesmium* spp.
is also characterized by the loss of buoyancy (collapse of gas vesicles) and increased
production of TEP and aggregation leading to enhanced and massive vertical flux (Bar-Zeev
et al., 2013). A *Trichodesmium* spp. bloom that occurred outside the VAHINE mesocosms on
days 23-24 displayed mechanistic features of PCD including mass mortality within 24 h, loss
of gas vesicles, and high production of TEP (Spungin et al., 2016). While we could not
directly quantify the export flux as no sediment traps were deployed in the lagoon water
outside the mesocosms, the characteristics of the bloom, lack of grazer influence and the
demise of biomass suggests this would lead to high rates of export (Spungin et al., 2016) as
demonstrated in culture simulations (Bar-Zeev et al., 2013) (Fig 5c).
**4.2.2 DDN transfer to zooplankton**
DDN transfer to zooplankton may either be direct through the ingestion of diazotrophs, or
indirect, i.e. mediated through the release of dissolved DDN by diazotrophs taken up by
heterotrophic and autotrophic plankton and subsequently grazed by zooplankton. During the
VAHINE experiment, the percent contribution of DDN to zooplankton biomass averaged 30



% (range = 15 to 70 %) (Hunt et al., 2016), which is in upper range of values reported from
high $N_2$ fixation areas such as the subtropical north Atlantic (Landrum et al., 2011; Mompean
et al., 2013; Montoya et al., 2002a), the Baltic Sea (Sommer et al., 2006; Wannicke et al.,
2013b), and the pelagic waters off the New Caledonian shelf (Hunt et al., 2015).
During VAHINE all four of the qPCR targeted diazotrophs (*Trichodesmium* spp., het-1, het-2,
UCYN-C) were found in zooplankton guts indicating a direct grazing of these four phylotypes
(Hunt et al., 2016). Overall, the most frequently detected targets were het-1 (during P1; 17 to
180 *nifH* copies copepod$^{-1}$) and UCYN-C (during P2; 7 to 50 *nifH* copies copepod$^{-1}$), i.e. the
most abundant phylotypes encountered in the mesocosms during P1 and P2, respectively.
However, *Trichodesmium* spp. and het-2 were also detected at relatively high abundances in
copepod guts (~280 *nifH* copies copepod$^{-1}$) despite their low abundance in the mesocosms,
suggesting selective feeding and a possible top down control through zooplankton grazing for
these two phylotypes.
Direct and efficient zooplankton grazing on UCYN-C was further substantiated by targeted
grazing experiments during VAHINE which consisted of $^{15}N_2$-labeled bottle incubations of
freshly collected zooplankton in the presence of natural phytoplankton assemblages. The $^{15}N_2$
label was taken up by the diazotroph in the incubation bottles and used as a marker of
zooplankton diazotroph ingestion and/or ingestion of non-diazotrophic plankton grown on
DDN. Zooplankton were highly $^{15}N$ enriched  after 72 h of incubation during the UCYN-C
bloom (P2), slightly enriched during P1 when DDAs dominated to diazotrophic community,
and not enriched at all when a *Trichodesmium* spp. bloom was encountered outside the
mesocosms during P2 (Hunt et al., 2016). This was a surprising finding given that het-1, and
to a lesser extent *Trichodesmium* spp. were detected in copepod guts, and would suggest that
UCYN-C are much more efficiently transferred to zooplankton compared to DDAs and
*Trichodesmium* spp. While we demonstrated direct grazing of zooplankton on *Trichodesmium*
spp., DDAs and UCYN-C, further studies are required to quantify a more general contribution
of direct and indirect transfer of DDN to zooplankton.

## 29  5 Modelling as a tool to infer the fate of DDN and the role of $N_2$ fixation on
## 30  productivity, food web structure and C export

Modelling has accompanied every stage of the VAHINE project. Mesocosm 1D-vertical
simulations with the biogeochemical mechanistic Eco3M-MED model (Alekseenko et al.,
2014), enriched with diazotrophs for the present study, and embedded in the Eco3M
modelling platform (Baklouti et al., 2006), were utilized prior to the *in situ* experiments to aid




in the scientific design of the experiment and in understanding the need and the optimal
timing of the DIP enrichment. The biogeochemical model was first assessed using *in situ* data
from the mesocosms and then applied to study the fate of DDN in the ecosystem (Gimenez et
al., 2016). Finally, one of the main strengths of the modelling tool lies in the opportunity that
it offers to deconvoluate the different processes that are deeply interlinked. This last facility is
used here to infer the role of $N_2$ fixation on productivity, food web structure and C export.
The simulation of the mesocosm experiment (including DIP enrichment) reported in Gimenez
et al. (2016) hereafter referred to as the 'REF' simulation, and its main results relative to the
fate of the DDN are summarized below.
At the end of the REF simulation (set at 25 days in the model), 33 % of the DDN was found
in the diazotrophs, 43 % in the non-diazotroph organisms, 16 % in the DON pool, 3 % in the
particulate detrital organic pool and 5 % in the traps, indicating that $N_2$ fixation efficiently
benefited non-diazotrophic organisms and contributed to particle export. The model results
substantiated the mass balance of N (Berthelot et al., 2015b) demonstrating  that during the 10
first days of the experiment, planktonic organisms did not significantly benefit from the DDN
and that DDN did not accumulate in the water column (was not transferred to non-
diazotrophic plankton). After day 10, the DDN proportion increased in all the non-
diazotrophic plankton groups, and simultaneously decreased in the non-living pools, although
DON concentrations lagged decreasing only from day 13. This decrease in DDN proportion in
the abiotic N pools is due both to the assimilation of mineral and organic nutrients by
phytoplankton and heterotrophic prokaryotes, as well as to the sinking of the produced
organic matter through aggregation processes.
The model results further showed that the fraction of DDN in the exported particulate matter
increased from day 10 until the end of the simulation, consistent with the high *e*-ratio
determined by (Berthelot et al., 2015b) during P2 (see above) and with the $\delta^{15}$N-budget
performed by Knapp et al. (submitted), emphasizing the higher contribution of $N_2$ fixation to
export production during P2 compared to P1 (Gimenez et al., 2016).
In the model, diazotrophs were assumed to release equal amounts of $NH_4^+$ and DON at a rate
which increases non-linearly with the absolute and relative N contents of diazotrophs
(Gimenez et al., 2016). During P1, DDN accumulated in the DON pool (nearly up to 40 % of
the DDN generated from the beginning of the experiment if found in DON on day 13),
whereas the proportion of DDN associated with $NH_4^+$ decreased rapidly from day 5 as  $NH_4^+$
was immediately used by heterotrophic bacteria and phytoplankton. The proportion of DDN




associated with DON decreased later (i.e. during P2) when the inorganic N pool was depleted.
The model results are consistent with the $^{15}$N measurements from the $NH_4^+$ and DON pools,
indicating that $NH_4^+$ was preferentially transferred to non-diazotrophic plankton compared to
DON, which accumulated in the dissolved pool (Berthelot et al., 2016).
The model results were further validated in the distribution of the DDN among the biotic
compartments. Small-size (pico- and nano-) phytoplankton, heterotrophic prokaryotes,
heterotrophic nanoflagellates and ciliates were the main beneficiaries of DDN, as observed by
the nanoSIMS studies (Berthelot et al., 2016; Bonnet et al., 2015a). Small-size phytoplankton
and heterotrophic prokaryotes were indeed the main consumers of $NH_4^+$ and labile DON (the
model excludes DON uptake by large-size phytoplankton), and heterotrophic nanoflagellates
and ciliates respectively feed on heterotrophic prokaryotes and small-size phytoplankton.
These results therefore indicate that DDN mainly transited through pico-, nanophytoplankton
and the actors of the microbial loop during the VAHINE experiment.
Both the *in situ* and modelling work summarized in the previous sections demonstrate the
important contribution and role of the diazotrophic communities to PP (non-diazotrophic) and
BP, to zooplankton feeding, and eventually to C export.
To further assess the role of $N_2$ fixation on the ecosystem, we used the REF simulation from
Gimenez et al. (2016) and compared it to a new simulation in which we removed the $N_2$
fixation capability of diazotrophs (hereafter named 'NOFIX simulation'). The NOFIX
simulation also included the following changes compared to the REF simulation to be
consistent with the new environmental conditions: (i) the initial relative N quotas of
diazotrophs have been set to 25 % (instead of 100 % in the reference simulation, i.e. same
value as the one used for non-diazotrophs). As the initial total N was identical to the one of
the REF simulation, the N content of diazotrophs has been allocated to the detrital N
compartment; (ii) all along the NOFIX simulation, only the detrital particulate compartment is
allowed to sink at a constant rate of 0.7 m d$^{-1}$ (see Gimenez et al. (2016)), whereas in the REF
simulation, this was also the case only until day 10 beyond which all the compartments were
allowed to sink at a rate increasing with time, in order to mimic the observed increase in the
particulate sinking flux due to TEP release and aggregation .
When comparing the REF and NOFIX simulations (Fig. 6), we note that the shapes of the PP
and BP curves remain the same, showing an increase in PP and PB during P2 in both
simulations. However, in the NOFIX simulation, the magnitude of PP and BP is reduced by




2.5 and 1.5-fold respectively. Furthermore, according to the model, $N_2$ fixation fueled 43.5 %
of PP and 8 % of BP during the 23 days of the simulated experiment.
The fact that the resulting PP was reduced to a larger extent than the BP when $N_2$ fixation was
absent did not necessarily mean that non-diazotrophic autotrophs benefit more from the DDN
compared to heterotrophs as the DDN was nearly equally distributed between autotrophs and
heterotrophs (and slightly higher in heterotrophs) (Gimenez et al., 2016). This higher effect on
PP than on BP is derived from the fact that the diazotrophs themselves (and therefore a part of
PP since only autotrophic diazotrophs were considered in the model) were strongly affected
by their inability to fix $N_2$ as suggested by the far lower abundance of UCYN-C in the NOFIX
simulation compared to the REF one (Fig. 6). This also explains why removing $N_2$ fixation
first affected PP (around day 10) compared to BP (around day 15).
We further assumed that, apart from diazotrophs, the organisms mostly influenced by the
absence of $N_2$ fixation (in the simulation) should be those organisms that benefited the most
from the DDN (i.e. in which the highest percentages of DDN have been calculated by the
model (see Fig. 6 in Gimenez et al. (2016)), namely small (< 10 μm) phytoplankton,
heterotrophic prokaryotes, heterotrophic nanoflagellates, and ciliates. This was the case for
small phytoplankton and heterotrophic bacteria (Fig. 7), and to a lesser extent and later for
heterotrophic nanoflagellates. This was also true for ciliate abundance, but only until day 16.
After day 16, ciliate abundance was slightly (<5 % between day 16 and 23) higher in the
NOFIX simulation compared to the REF one, resulting predominantly from a top-down effect
due to increased copepod predation in the NOFIX simulation from day 10 to day 23 (results
not shown).
Our model did not include DDAs and did not allow the uptake of DON by large
phytoplankton (i.e. diatoms). Thus, the DDN content in diatoms, and therefore in
mesozooplankton, was probably slightly underestimated by the model in the REF simulation
(Gimenez et al., 2016) compared to *in situ* data (Hunt et al., 2016). As a result, large
phytoplankton and mesozooplankton abundances were nearly similar in the REF and NOFIX
simulations (not shown). Hence, apart from ciliates (which mortality also fuels the detrital
particulate compartment as for large phytoplankton and mesozooplankton), the organisms that
mostly benefited from the DDN were small organisms, the mortality of which fuels the
dissolved organic pool.
How does $N_2$ fixation impact C export? Absence of $N_2$ fixation (NOFIX simulation) reduced
export by 30 % on day 23 compared to the REF simulation (Fig. 8). This difference in C




export reaches 50% when the simulation duration is extended until day 35 (not shown). These
results indicate that $N_2$ fixation and the subsequent new production promotes C export to
depth as the experimental VAHINE results demonstrated (Berthelot et al., 2015b; Knapp et
al., 2015).
A third simulation (not shown) in which the $N_2$ fixation capability by diazotrophs is still
removed but in which the aggregation processes were represented (in the same way as in the
REF simulation) indicated that C export is nearly equal to that of the REF simulation after 25
days (they differ by only 2.9 %), with 25 % difference reached on day 35. This suggests that
the higher C export when $N_2$ fixation is active occurs initially due to aggregation processes
mediated diazotrophs-derived TEP release and the subsequent export of diazotrophs (Berman-
Frank et al., 2016; Bonnet et al., 2015a). Moreover, it is likely that increased stickiness and
aggregate properties also cause further accumulation, aggregation, and enhanced vertical flux
from the different compartments in the water column. To represent the latter phenomenon, we
considered that 10 % of the living and non-living compartments are allowed to sink after day
10 in the model (see Gimenez et al. (2016) for more details). In a second step however, the $N_2$
fixation process per se (by supporting PP and BP fluxes) contributes more and more to the
enhanced C export as $N_2$ fixation fluxes increase. Hence, on day 30, $N_2$ fixation supports ~50
% of the excess C export observed between the REF and the NOFIX simulations, the
remaining still being attributed to aggregation processes.
To conclude, $N_2$ fixation has a significant impact on both direct and indirect C export via
diazotroph fueling of non-diazotrophic plankton as well as via aggregation processes. The
model provides a lower limit of the major role played by $N_2$ fixation on C export due to an
underestimate of the DDN content in diatoms, and in mesozooplankton. Finally, this study
also points the need of further investigation on aggregation processes in relation with TEP
release and its representation in models since its influence on C export may be of the same
order of magnitude as the $N_2$ fixation process per se.

## 6    Conclusions and future work

The VAHINE project provided unique opportunities to study and compare the fate of $N_2$
fixation associated with different diazotrophs in the marine environment. The results showed
that when the diazotroph community was dominated by DDAs, the DDN remained within the
symbiotic associations, was poorly transferred to the non-diazotrophic phytoplankton and
heterotrophic prokaryotes, yet can be transferred directly to zooplankton through grazing. The
project results further substantiated previous data showing rapid export to depth of the



recently fixed $N_2$ by DDAs (Karl et al., 2012). An opportune bloom of UCYN-C during the
VAHINE project demonstrated that when UCYN-C dominated the diazotroph community, ~
25 % of the DDN was quickly (24 h) transferred to the planktonic food web through the
release of DON and $NH_4^+$ to the dissolved pool. These additional N sources were
subsequently transferred to zooplankton, both directly (through the grazing of UCYN-C) and
indirectly through the grazing of plankton grown on DDN from UCYN-C. Moreover, the
VAHINE data explicitly revealed that when UCYN-C dominate the diazotroph community,
the efficiency of the system to export POC relative to PP (*e*-ratio) is higher than when DDAs
dominate. This export is both direct through the sinking of small (5.7±0.8 μm) UCYN-C cells
aggregated into large (100-500 μm) particles having high sinking rates, and indirect through
the sinking of plankton benefitting from the enriched source of DDN. Future projects should
extend the investigation of DDN export below the photic layer in the open ocean (~70-150 m
in the oligotrophic ocean) to confirm the process study obtained during VAHINE in
mesocosms in an experimental 15 m-depth water column. In particular, are the aggregation
processes of UCYN also observed in the open ocean? Although technically and logistically
challenging, this feat may be accomplished through locating a research vessel in a 1D
structure (cyclonic eddy harboring high UCYN abundances for example) where horizontal
advection is reduced and sediment traps are deployed to study the biological and
biogeochemical characteristics of the photic zone for one to two weeks.
The VAHINE project also provided a unique opportunity to compare the transfer efficiency of
DDN from UCYN and *Trichodesmium* spp. to the different compartments of the planktonic
food web, and revealed that the main beneficiaries of the DDN depend on both the
physiological status (e.g. nutritionally balanced, stationary or decline phase) and the type of
diazotroph. When *Trichodesmium* spp. bloom decay they release large amounts of $NH_4^+$ and
mainly support diatom growth, indicating a large potential of indirect organic matter export
during/after *Trichodesmium* spp. blooms. This is further substantiated by the study of PCD
indicating a rapid direct export of *Trichodesmium* spp. itself but further studies are needed in
open ocean *Trichodesmium* spp. blooms to extrapolate our results to the field.
$NH_4^+$ appears to be the main form of DDN transferred to non-diazotrophic plankton. In future
studies, it would be necessary to refine the chemical composition of DON released by
different diazotrophs to assess its lability as a function of the diazotrophs involved in $N_2$
fixation and the stage of the bloom. It would also be informative to explore the amount and
chemical composition of released DOC and better study the potential of diazotrophs to
stimulate heterotrophs and their subsequent impact on the ocean metabolic balance.





Finally, in the future ocean, some diazotrophs such as *Trichodesmium* spp. (Hutchins et al.,
2007; Levitan et al., 2007) and UCYN-B (Fu et al., 2008) (no study is available on UCYN-C)
may develop extensively under high temperature and $p\mathrm{CO_2}$ conditions (Dutkiewicz et al.,
2015), while other such as UCYN-A would not be affected (Law et al., 2012). The results
from the VAHINE project revealed that the diazotroph community composition has a
profound impact in structuring the planktonic food web in the surface ocean, and in the
efficiency of particulate matter export to depth. Thus, current and predicted global changes
require further knowledge and understanding of the fate and implications of changing
scenarios of $\mathrm{N_2}$ fixation in the future oceans.
**Acknowledgements**
Funding for this research was provided by the Agence Nationale de la Recherche (ANR
starting grant VAHINE ANR-13-JS06-0002), the INSU-LEFE-CYBER program, GOPS and
IRD. The authors thank the captain and crew of the R/V *Alis*. We acknowledge the SEOH
diver service from Noumea, as well as the technical service of the IRD research center of
Noumea for their helpful technical support together with C. Guieu, J.-M. Grisoni and F. Louis
for the mesocosm design and the useful advice. Partial funding to IBF was provided through a
collaborative grant with SB from MOST Israel and the High Council for Science and
Technology (HCST)-France, and a GIF grant No. 1133-13.8/2011.


**Figure legends.**
**Figure 1.** Study site of the VAHINE experiment. Location map of New Caledonia in the
Southwestern Pacific (a), Map of the Noumea lagoon showing the location of mesocosms at
the entrance of the lagoon, 28 km off the coast (b).
**Figure 2.** View of the mesocosms from above (a), from the seafloor (b) and view of the
sediment traps that collect  sinking particles (c) (Photos credits: J.M. Boré and E. Folcher,
IRD).
**Figure 3.** Evolution of sea surface temperature (°C) (a), $NO_3^-$ ($\mu$mol L$^{-1}$) (b), DIP ($\mu$mol L$^{-1}$)
(c), Chl a ($\mu$g L$^{-1}$) (d), $N_2$ fixation rates (nmol N L$^{-1}$ d$^{-1}$) (e), PON concentrations ($\mu$mol L$^{-1}$)
(f), DON concentrations ($\mu$mol L$^{-1}$) (g) and PON export ($\mu$mol L$^{-1}$) (h) over the 23 days of the
VAHINE mesocosm experiment. Lines represent the average of the three mesocoms and
shaded areas represent the measured min and max values.
**Figure 4.** Upper panel: Diazotroph community composition in the VAHINE mesocosm
experiment during the experimental period. *nifH*-based abundances were summed for each
sampling day to determine the percent contribution to the total diazotroph community from
each major phylotype (data from Turk-Kubo et al. (2015)). Bottom panel: simplified
evolution of the major standing stocks, rates and plankton abundances measured during P1
(days 5 to 14) and P2 (days 15 to 23) in the mesocosms. Squares are represented in green
when a significant (p<0.05) increase was observed between each period (i.e. between P0 and
P1 or between P1 and P2, Kruskall-Wallis test, $\alpha$=0.05), in red when a significant (p<0.05)
decrease was observed and in grey when no significant change was observed between the
different periods.
**Figure 5.** Summary of the simplified pathways of the potential DDN transfer in the first
trophic level of the food web and potential of direct *versus* indirect export of particulate
matter for DDAs (a), UCYN-C (b) and *Trichodesmium* (c). DDN transfer data from (Bonnet
et al., Accepted; Bonnet et al., 2015a)



**Figure 6.** Evolution of PP ($\mu$mol C L$^{-1}$ d$^{-1}$) (a) and bacterial production (ng C L$^{-1}$ h$^{-1}$) in the
REF simulation (black line) and the NOFIX simulation (blue line) (i.e. when the N$_2$ fixation
process is removed).
**Figure 7.** Evolution of plankton abundances (cells L$^{-1}$) in the REF simulation (black line) and
the NOFIX simulation (blue line) (i.e. when the N$_2$ fixation process is removed). TRI:
*Trichodesmium* spp., UCYN: UCYN-C, BAC: heterotrophic bacteria, PHYS: small
phytoplankton, HNF: heterotrophic nanoflagellates.
**Figure 8.** Evolution of C content collected in the mesocosm particle traps (mmol C) in the
REF simulation (black line) and the NOFIX simulation (blue line) (i.e. when the N$_2$ fixation
process is removed).



a)                      b)

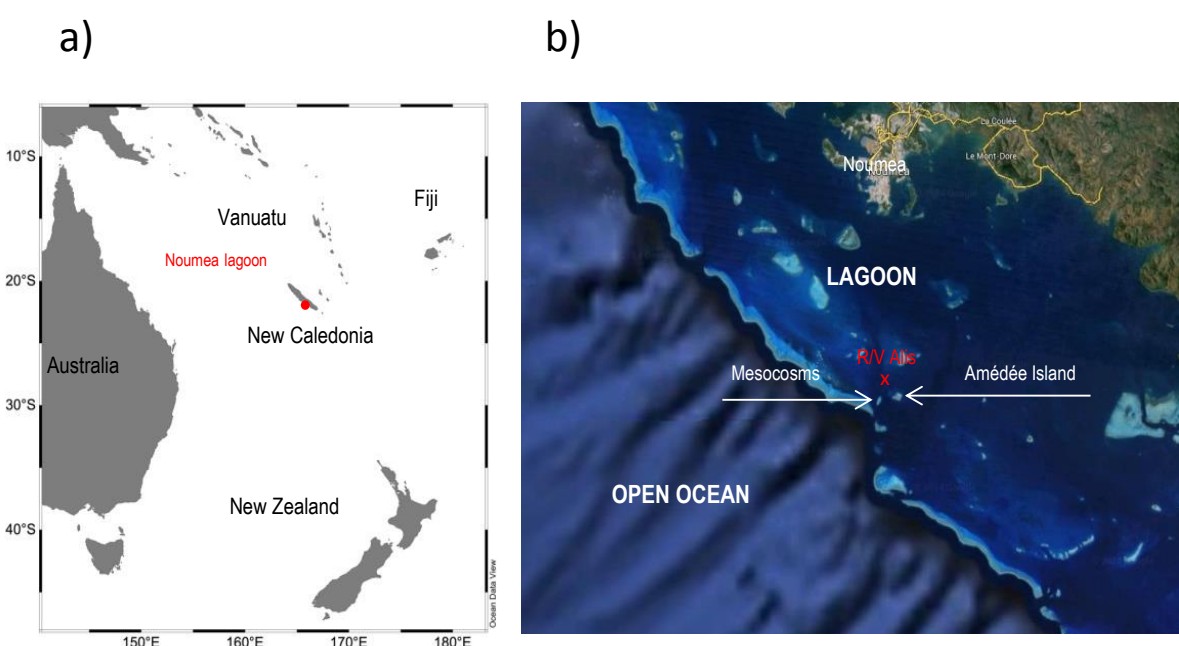

Figure 1.



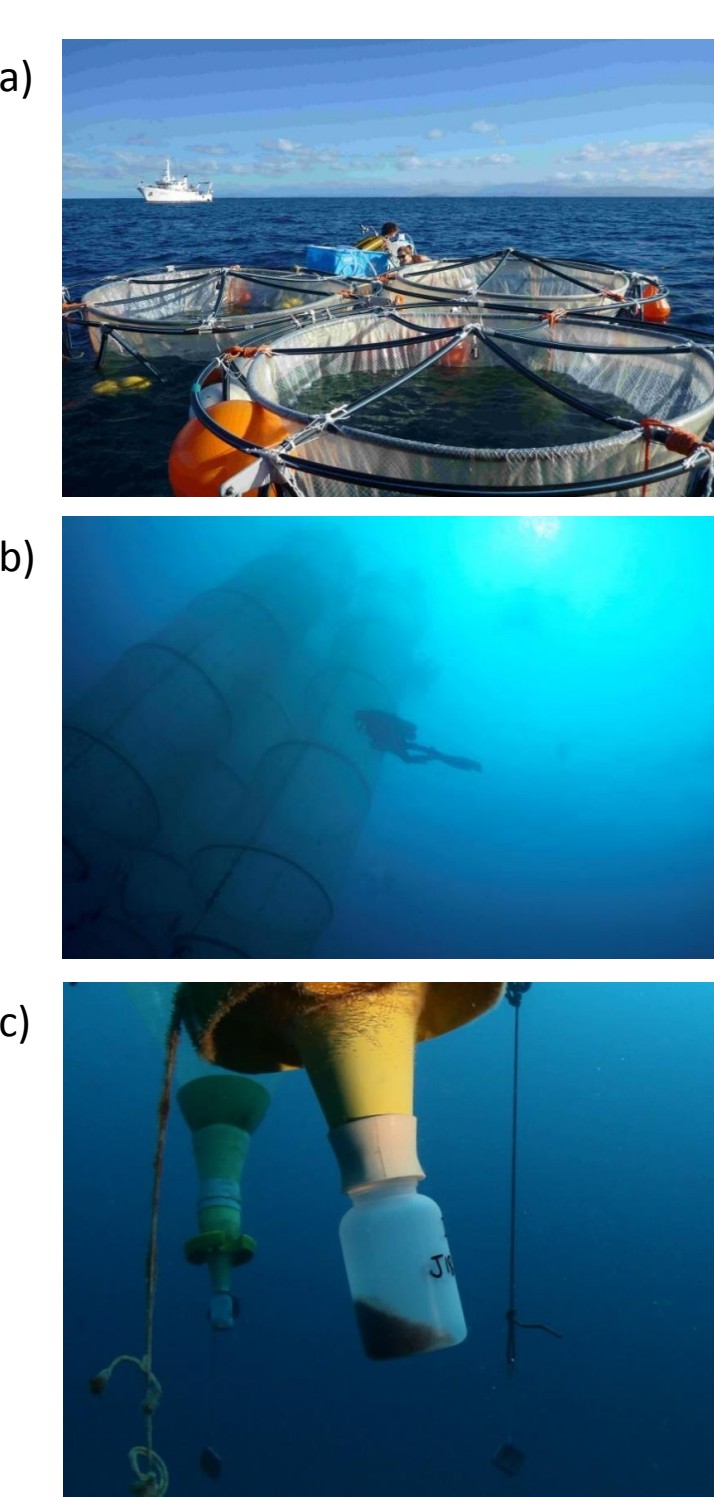

Figure 2.

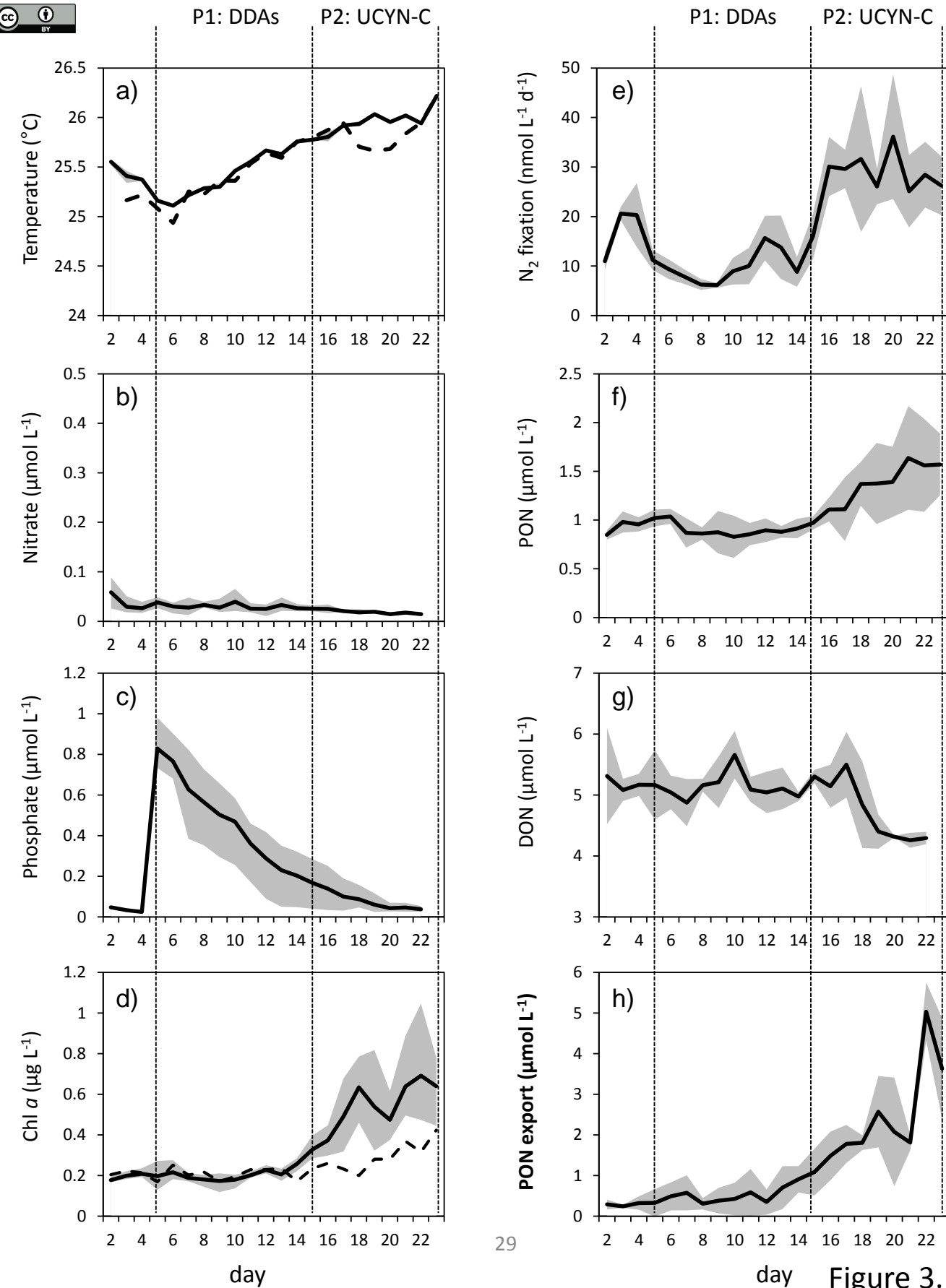

Figure 3.

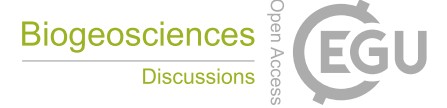

Figure 4.





Figure 5.





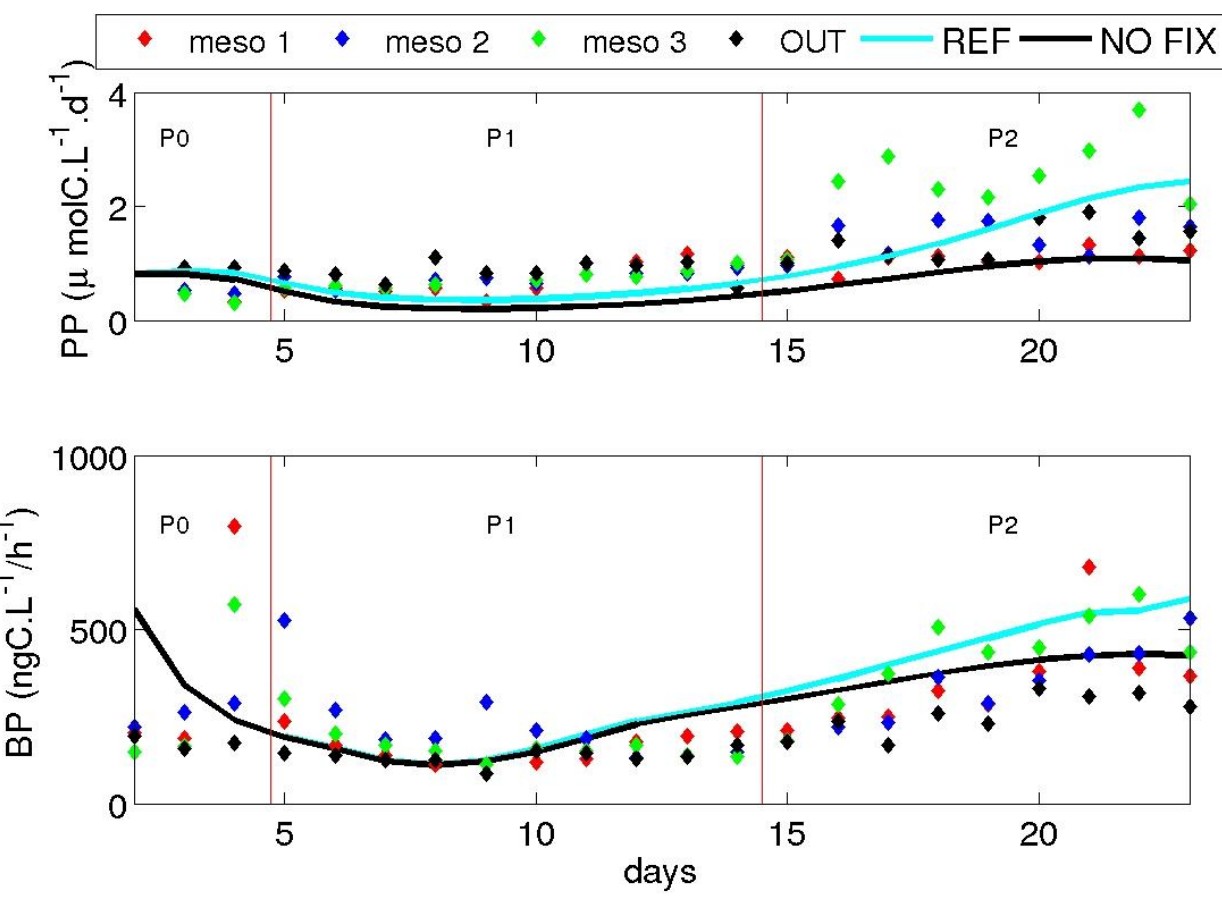

Figure 6.





Figure 7.





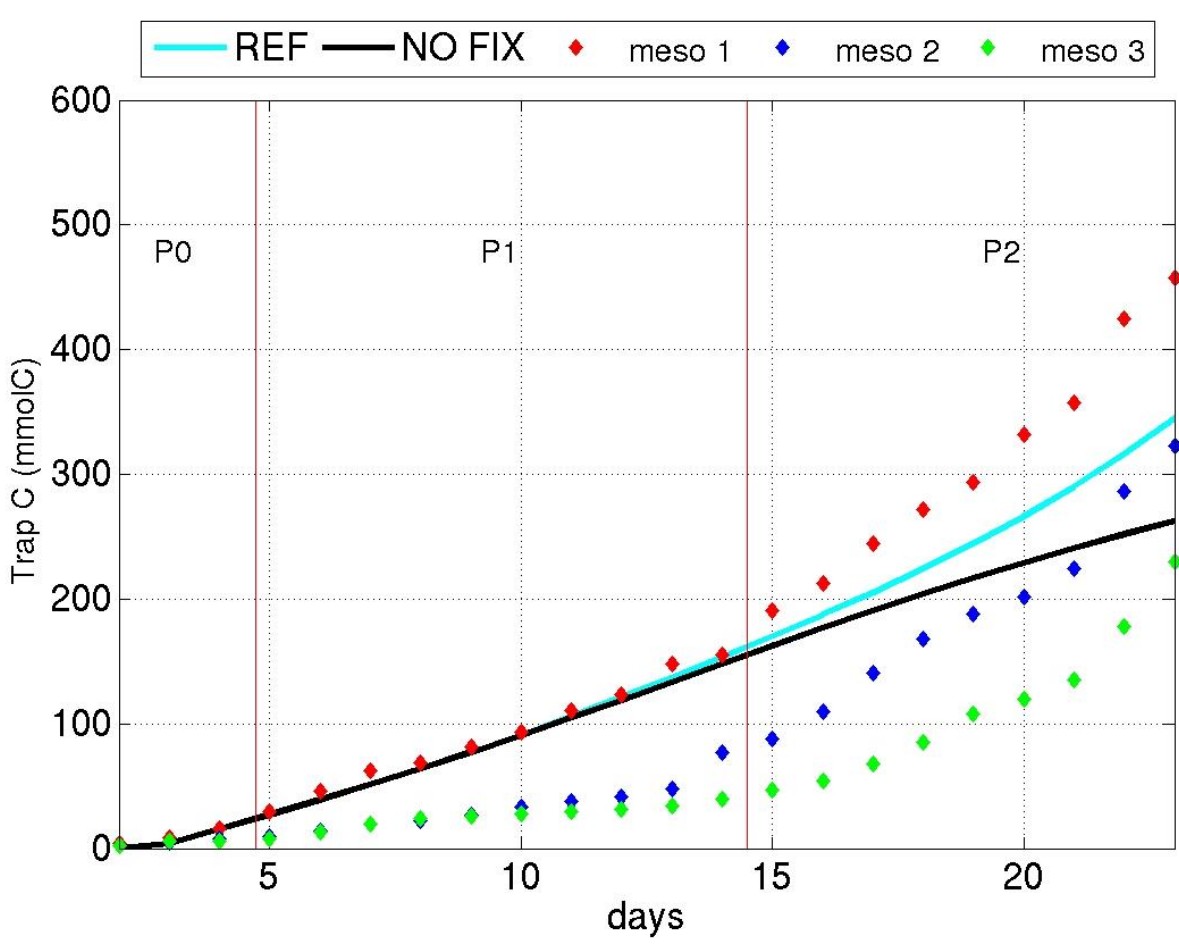

Figure 8.





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
