# Peer review of "Biogeochemical and biological impacts of diazotroph"

_Biogeosciences, 2015_

## Referee Comment (RC1) · J. Corredor (Referee) · 20 Mar 2016

J. Corredor (Referee)

jorge.corredor@upr.edu

bg-2015-668 reviewJC GENERAL COMMENTS This paper constitutes an overview of the results of the VAHINE project; a multidisciplinary mesocosm field experiment designed to elucidate transport mechanisms and eventual fate of nitrogen fixed by diazotrophs in the tropical ocean surface following phosphorus addition. This is a question of great current interest to allow further insight into the role of this process in fueling biogeochemical turnover over great expanses of the world ocean where phosphorus availability is limiting to phytoplankton growth. In the context of global change, it can

provide better understanding of the role of N cycling in C sequestration in the deep ocean. Overall, this is a well written manuscript that succinctly conveys the salient results of this complex experiment. SPECIFIC COMMENTS p.16: 19-20 – Authors argue that detritus and DON . . . "likely provided" the balance of bacterial N demand unaccounted by DDN since concentrations of these two components "decreased during the 23 days of the experiment". A couple of rapid calculations can easily dispel this doubt providing a better view of relative magnitudes and revealing gaps in this budget if at all. p. 17:7-28 - The review of the role of Trichodesmium in N export, while pertinent to the discussion, is beyond the scope of the mesocosm experiment and should thus be abbreviated considerably

TECHNICAL CORRECTIONS Abstract: 9 – delete "potential" p.7 11-23 – It is preferable to pose your objectives as statements rather than questions. Indeed, objective iii is posed at a statement but provided with an erroneous question mark! The same for the first line in objective iv. 27 – consider changing "stable" for "unique" p.8 14 – change "has been" to "was". (The experiment is not ongoing; it was terminanted after 23 days). 16 change "harbouring" to "exhibiting" p.13 21 – refrain from citing your work as the "first". If it really is, others will identify it as such. 26 – Change "way" to "pathway" p.14 1- Change "the one" to "that" 15- Change "The export" to "Export"; change "has not" to "was" 26 – Rephrase to eliminate innapropriate question mark. 28 – Eliminate redundancy; change second "UCYN-C" to "these" p.6 15 – Rephrase to avoid "first". Perhaps, "We thus demonstrate that UCYN blooms may result in substantial DDN release."

---

## Referee Comment (RC2) · Anonymous Referee #2 · 24 Mar 2016

Biogeochemical and biological impacts of diazotroph blooms in a Low Nutrient Low Chlorophyll ecosystem: synthesis from the VAHINE mesocosm experiment (New Caledonia)

Sophie Bonnet et al.

This manuscript provides a generally well written summary of the VAHINE mesocosm experiment which informs on current interests in the contribution of diazotrophs to carbon export and their support of other microbial communities and higher trophic levels through the production of bioavailable nitrogen. My suggestions are mostly confined

to the presentation of the written work. There are a number of occasions where the language used is a little clumsy and I have made some suggested edits. As currently written the manuscript assumes a reasonable amount of prior knowledge of the subject area and of this experiment in particular. I would advise better definition of terms and referencing of other papers which support this synthesis – a number of which I have highlighted.

P2L11, P7L28 my experience suggests that mesocosms do disturb the ambient light field. Please provide evidence to the contrary or reference to the relevant paper in the special issue to support this.

P3L12 tropical LNLC ecosystems include . . ..subtropical gyres? Tropical and subtropical are different environments/regimes

P3L22 need an extra ) after 2008))

P3L29 . . . . . ...cycles HAS been

P3L32 preferentially exported directly . . .. . ..

P5L12 phytoplankton. Actual Calculations of DDN transfer were first . . ...

P5L21 . . .poorly and challenged qualified due mainly to

P9L4 ..good replicability low variability

P9L18 within MESOCOSMS and in . . ...

P11L2 How is DIP turn-over time defined and measured. I do not see any reference to DIP uptake rates. This needs some detail or referencing to the appropriate paper.

P11 A number of rate measurements are introduced on this page which do not seem to be defined (e.g. APA, PP, BP) neither is there reference to the papers containing this data/description.

P12L5-8 How do your results demonstrate this? No evidence is given here neither is

there any reference to the paper detailing this work.

P12L28-29 . . ...to determine whether . . . .. . .of particulate matter, and if so, how was this manifested.

P12L34 equalled

P13L27 what is nanosims and how was it used to demonstrate this?

P14L2 define e-ratio

P14L28-32 long sentence which needs breaking up

P15L1 Are waters contained within a mesocosm natural?

P15L23 Surely the evidence to date indicates that the bubble method underestimates rates? (Mohr et al, Grosskopf et al, etc)

P16L22-27 long sentence

P17L10-15 long sentence

P18L10-13 Whilst Trichodesmium is not a particular focus of this manuscript, the finding that copepods are potentially selectively feeding on Tricho warrants further investigation.

P19L5 deconvoluate???? No idea what is meant by this

P19L15-16 .. during the first 10 days . . .

P19L31-32 (nearly up to 40% of the DDN . . . .. .. . ..experiment is found . . .)

P20L12 DDN was mainly transferred through . . .

Figs 6,7,8 Labelling of REF and NOFIX in figure legends is the wrong way round

P22L9-10 aggregation processes mediated diazotrophs-derived TEP release – this needs re-phrasing somehow

P23L26 what is PCD?

---

## Referee Comment (RC3) · Anonymous Referee #3 · 5 Apr 2016

The interdisciplinary VAHINE project has already generated a large number of data-rich papers, a dozen of which are cited in this paper. This current manuscript provides a summary (synthesis) of some of the major trends from this controlled mesocosm experiment. I have not gone back and read all the individual papers so I cannot really comment on the accuracy or inclusive nature of this summary; hence, I do not have an informed opinion of whether it is needed as a "stand alone" paper. I was surprised to learn that yet another paper (listed in the reference list as Bonnet et al., in preparation) termed "Introduction to the project VAHINE" is planned. It struck me as odd that no "introduction" had yet been published, given the many papers that have already appeared. Why not combine the introduction and the synthesis into a single paper? That would seem logical to this reader.

Specific Comments

p. 2, line 11: "a stable water mass" – Was turbulence measured?

p. 3, line 5: ammonia is NH3, ammonium is NH4+

p. 3, line 6: crops, not cultures?

p. 5, line 21: quantified, not qualified?

p. 6, line 22: Eastern Tropical Pacific?

p. 8, line 17: 40 nM NO3- seems high to me. So does 0.1-0.15 ïĄ■g Chl a l-1

Fig. 3: Why not plot particulate P and DOP?

Fig. 3: units on (h) PON export seem to be incorrect

---

## Author Response (AR1)

Dear Referees,

First, we would like to thank you very much for your constructive comments. We are pleased to provide a revised version of our manuscript *"Biogeochemical and biological impacts of diazotroph blooms in a Low Nutrient Low Chlorophyll ecosystem: synthesis from the VAHINE mesocosm experiment (New Caledonia)"*. We made our best to take into consideration all comments and suggestions. Comments and questions are in regular font with our replies below in bold font. The marked-up manuscript is also provided below.

Sophie Bonnet on behalf of co authors

Referee #1

SPECIFIC COMMENTS

p.16: 19-20 – Authors argue that detritus and DON … "likely provided" the balance of bacterial N demand unaccounted by DDN since concentrations of these two components "decreased during the 23 days of the experiment". A couple of rapid calculations can easily dispel this doubt providing a better view of relative magnitudes and revealing gaps in this budget if at all.

**The sentence has been modified as follows: "Calculations based on C:N molar ratios show that $N_2$ fixation may have provided ~30 % of the N demand of the N-limited bacteria during P2 (compared to ~20 % during P1), the rest (representing 0.6-0.7 µmol $L^{-1}$) was likely provided by detritus and DON (Van Wambeke et al., 2015), which concentrations decreased by ~0.9 µmol $L^{-1}$ during the 23 days (Berthelot et al., 2015b)."**

p. 17:7-28 - The review of the role of *Trichodesmium* in N export, while pertinent to the discussion, is beyond the scope of the mesocosm experiment and should thus be abbreviated considerably

**A *Trichodesmium* bloom occurred during the VAHINE experiment, albeit outside the mesocosms. Nevertheless, this bloom has been characterized and results are presented in Spungin et al. (2016) in the present special issue. We strongly believe that these results are worth considering in the present synthesis article. Nevertheless, we considerably reduced this section as follows:**
**"Similar experiments ($^{15}N_2$ labelling, flow cytometry cell sorting and nanoSIMS) performed on three naturally-occurring *Trichodesmium* spp. blooms in the southwestern Pacific illustrated that DDN was predominantly transferred to diatoms (Bonnet et al., Accepted). These results indicate that the extensive oceanic blooms of *Trichodesmium* spp. can contribute to a subsequent indirect yet large downward flux of organic matter by promoting large cells growth (e.g., diatoms and dinoflagellates) characterized by efficient export rates (Nelson et al., 1995, Bonnet et al., Accepted; Devassy et al., 1979; Lenes et al., 2001).**
**Direct export flux of *Trichodesmium* spp. blooms may also occur in cases where rapid (< 2 d) bloom mortality occurs via a programmed cell death (PCD) (Berman-Frank et al., 2004;**

**Berman-Frank et al., 2007). PCD in *Trichodesmium* spp. is characterized by the loss of buoyancy (collapse of gas vesicles) and increased production of TEP and aggregation leading to enhanced and massive vertical flux (Bar-Zeev et al., 2013). A *Trichodesmium* spp. bloom that occurred outside the VAHINE mesocosms on days 23-24 displayed mechanistic features of PCD including mass mortality within 24 h, loss of gas vesicles, and high production of TEP (Berman-Frank et al., 2016; Spungin et al., 2016). While we could not directly quantify the export flux as no sediment traps were deployed in the lagoon outside the mesocosms, the characteristics of the bloom, minimal grazer influence and the demise of biomass suggests this would lead to high rates of export (Spungin et al., 2016) as demonstrated in culture simulations (Bar-Zeev et al., 2013) (Fig 5c). "**

TECHNICAL CORRECTIONS
Abstract: 9 – delete "potential"

**This word has been deleted**

p.7 11-23 – It is preferable to pose your objectives as statements rather than questions. Indeed, objective iii is posed at a statement but provided with an erroneous question mark! The same for the first line in objective iv.

**The objectives have been reformulated as statements:**
**The main scientific research priorities of the project were:**
**i)      To quantify the DDN which enters the planktonic food web,**
**ii)     To investigate how the development of diazotrophs influences the subsequent diversity, gene expression, and production of primary producers, heterotrophic bacterioplankton, and subsequently zooplankton abundance,**
**iii)    To examine whether different functional types of diazotrophs significantly modify the stocks and fluxes of the major biogenic elements (C, N, P),**
**iv)     To elucidate whether the efficiency of particulate matter export depends on the development of different functional types of diazotrophs.**

– consider changing "stable" for "unique"
p.8 14 – change "has been" to "was". (The experiment is not ongoing; it was terminated after 23 days).
change "harbouring" to "exhibiting"

**These changes have been applied.**

p.13 21 – refrain from citing your work as the "first". If it really is, others will identify it as such.

**We concur and have changed this especially on page 15.**

– Change "way" to "pathway"
p.14 1- Change "the one" to "that"
15- Change "The export" to "Export"; change "has not" to "was"
– Rephrase to eliminate inappropriate question mark.

28 – Eliminate redundancy; change second "UCYN-C" to "these"
p.15 6 – Rephrase to avoid "first". Perhaps, "We thus demonstrate that UCYN blooms may
result in substantial DDN release."

**The suggested changes have been applied.**

Referee #2

P2L11, P7L28 my experience suggests that mesocosms do disturb the ambient light
field. Please provide evidence to the contrary or reference to the relevant paper in the
special issue to support this.

**We agree that the mesocosms may have produced a slight change in the light**
**environment, albeit this was probably limited. So, we modified the text as follows in the**
**revised version of the manuscript: "….to maintain a stable water-mass minimizing the**
**disturbance of ambient light and temperature conditions…". In addition, we added the**
**Guieu et al. (2010) reference that describe extensively the mesocosms setup used during**
**the VAHINE experiment and where the question has been more largely discussed.**

P3L12 tropical LNLC ecosystems include : : :.subtropical gyres? Tropical and subtropical
are different environments/regimes

**We agree with this comment and removed "Tropical" at the beginning of the sentence.**

P3L22 need an extra ) after 2008))
P3L29 ……cycles HAS been
P3L32 preferentially exported directly …..
P5L12 phytoplankton. Actual Calculations of DDN transfer were first ……

**All these suggested changes have been done**

P5L21 : …poorly and challenged qualified due mainly to

**Rephrased as follows: "remains poorly qualified and challenging"**

P9L4 ..good replicability low variability

**Rephrased as follows: "These studies also revealed a good replicability and low variability**
**between stocks, fluxes and plankton diversity measurements among the replicate**
**mesocosms. …**

P9L18 within MESOCOSMS and in ….

**The word addition has been applied in the new version of the manuscript.**

P11L2 How is DIP turn-over time defined and measured. I do not see any reference to

DIP uptake rates. This needs some detail or referencing to the appropriate paper.
**DIP turn over time was measured using the radioisotope $^{33}$P according to Duhamel et al.**
**(2006). Results and details on DIP turn-over time measurements during the VAHINE**
**experiment are presented in Berthelot et al. (2015). The references for each of the**
**parameters have been added in the caption of Figure 4.**
P11 A number of rate measurements are introduced on this page which do not seem
to be defined (e.g. APA, PP, BP) neither is there reference to the papers containing this
data/description.
**All acronyms are now defined in the text when they appear for the first time.**
P12L5-8 How do your results demonstrate this? No evidence is given here neither is there
any reference to the paper detailing this work.
**This is detailed in the Gimenez et al paper within the special issue as cited in the text.**
**However, we acknowledge that this part is not the main goal of the present paper and we**
**decided to remove this paragraph. .....**
P12L28-29 : : :..to determine whether : : :: : :.of particulate matter, and if so, how was
this manifested.
**The suggested change has been applied.**
P12L34 equalled
**The suggested change has been applied.**
P13L27 what is nanosims and how was it used to demonstrate this?
**NanoSIMS refers to nanoscale Secondary Ion Mass Spectroscopy. The sentence has been**
**modified as follows: 'An experiment performed during the UCYNC bloom using nanoSIMS**
**(nanoscale Secondary Ion Mass Spectroscopy) as described in Bonnet et al., (2016)**
**demonstrated that a significant…'**
P14L2 define e-ratio
**The e-ratio depict the efficiency of the carbon export compared to primary production. In**
**order to clarify, we modified the text as follows: "indicated by e-ratio calculations (e-ratio**
**= PP/POC$_{export}$), which quantify the efficiency of a system to export particulate C relative to**
**the C fixed by PP).**
P14L28-32 long sentence which needs breaking up
**The sentence has been divided in two as follows: "During the maximal abundance of**
**UCYN-C, these were responsible for 90±29 % of total N$_2$ fixation rates in the mesocosms**

**(Bonnet et al., 2015a). During this period, the DDN released to the dissolved pool (based**
**on the direct measurement of the isotopic signature ($^{15}$N) of the total dissolved N**
**according to the denitrifying method (Knapp et al., 2005)) accounted for 7.1±1.2 to**
**20.6±8.1 % of gross N$_2$ fixation (Bonnet et al., 2015a).”**
P15L1 Are waters contained within a mesocosm natural?
**The waters present in the mesocosms were isolated from the lagoon the first day of the**
**experiment. The mesocosms are designed to minimize the perturbations (temperature,**
**light…) and reproduce as much as possible the natural environmental conditions.**
**Nevertheless, we agree that in the strict mean of the term, “natural” can be seen as**
**inappropriate and we deleted it in the new version of the manuscript.**
P15L23 Surely the evidence to date indicates that the bubble method underestimates
rates? (Mohr et al, Grosskopf et al, etc)
**Indeed, Mohr et al. 2010 reference appears to be more suitable in the context of the**
**sentence and replace the Montoya et al. 1996 reference in the new version of the**
**manuscript.**
P16L22-27 long sentence
**The sentence has been divided in two in the new version of the manuscript: “The**
**relationships between BP and N$_2$ fixation rates were weak (during P2) or absent (during**
**P1) but tightly coupled between BP and Chl *a* concentrations, and between BP and PP. This**
**suggests that N$_2$ fixation stimulated autotrophic communities and these subsequently**
**stimulated heterotrophic prokaryotes through the production and release of dissolved**
**organic matter including C (DOC) (Van Wambeke et al., 2015).”**
P17L10-15 long sentence
**The sentence has been modified as follows: ”These results indicate that the extensive**
**oceanic blooms of *Trichodesmium* spp. can contribute to a large indirect downward flux of**
**organic matter by promoting large cells (e.g., diatoms and dinoflagellates) characterized by**
**efficient export rates (Nelson et al., 1995, Bonnet et al., Accepted; Devassy et al., 1979;**
**Lenes et al., 2001).”**
P19L5 deconvoluate???? No idea what is meant by this
**The term “deconvoluate” has been changed by “separate”.**
P19L15-16 .. during the first 10 days ….
P19L31-32 (nearly up to 40% of the DDN ……experiment is found …..)
P20L12 DDN was mainly transferred through ……
**The suggested change has been applied in the new version of the manuscript.**

Figs 6,7,8 Labelling of REF and NOFIX in figure legends is the wrong way round
**The captions of Figs. 6,7 and 8 have been corrected in order to fix this problem.**
P22L9-10 aggregation processes mediated diazotrophs-derived TEP release – this
needs re-phrasing somehow
**The sentence has been rephrased and the paragraph in question has been reorganized to**
**improve the clarity of the text :**
**"It is likely that during the experiment, TEP release favored aggregation and accumulation**
**of particles and subsequently enhanced vertical flux from the different compartments in**
**the water column. To represent the latter phenomenon, we considered in the model that**
**10 % of the living and non-living compartments were allowed to sink after day 10 (see**
**Gimenez et al. (2016) for more details). Since this extra aggregation is mainly attributable**
**to diazotrophs, it was not represented in the NOFIX simulation. However, we ran a third**
**simulation (not shown) to further analyze the excess of C export in the REF simulation as**
**compared to the NOFIX one (Fig. 8). This third simulation is intermediate between the REF**
**and the NOFIX simulations in that sense that only the $N_2$ fixation capability by diazotrophs**
**is removed (but aggregation processes are still represented). This simulation indicated that**
**C export is nearly equal to that of the REF simulation after 25 days (they differ by only 2.9**
**%), thereby suggesting that during the 25 first days, the suppression of N2 fixation does**
**not significantly impact carbon export fluxes. This further suggests that the higher C export**
**in the REF simulation during P2 (Fig.8) is mainly due to aggregation processes mediated by**
**diazotrophs-derived TEP release and the subsequent export of diazotrophs (Berman-Frank**
**et al., 2016; Bonnet et al., 2015a). However, beyond day 25, the difference in carbon**
**export between the REF and the third simulation increases up to 25% on day 35. In other**
**words, the $N_2$ fixation process per se (by supporting PP and BP fluxes) contributes more**
**and more to the enhanced C export as $N_2$ fixation fluxes increase. Hence, on day 30, $N_2$**
**fixation supports ~50 % of the excess C export observed between the REF and the NOFIX**
**simulations, the remaining still being attributed to aggregation processes".**
P23L26 what is PCD?
**PCD means "Programmed Cell Death" and is defined in the new version of the manuscript.**
Referee #3
The interdisciplinary VAHINE project has already generated a large number of data rich
papers, a dozen of which are cited in this paper. This current manuscript provides a summary
(synthesis) of some of the major trends from this controlled mesocosm experiment. I have
not gone back and read all the individual papers so I cannot really comment on the accuracy
or inclusive nature of this summary; hence, I do not have an informed opinion of whether it
is needed as a "stand alone" paper. I was surprised to learn that yet another paper (listed in
the reference list as Bonnet et al., in preparation) termed "Introduction to the project
VAHINE" is planned. It struck me as odd that no "introduction" had yet been published,
given the many papers that have already appeared. Why not combine the introduction and
the synthesis into a single paper? That would seem logical to this reader.

**Actually the Introductory paper is already published in BG discussion (http://www.biogeosciences-discuss.net/bg-2015-615/) and has been recently accepted for final publication in BG after minor revisions. We agree that it was misleading as it appeared as 'in prep' in the present paper.**

**This intro paper aims at describing the scientific objectives of the project as well as the implementation plan: the mesocosms description and deployment, the selection of the study site (New Caledonian lagoon) and the logistical and sampling strategy. The main hydrological and biogeochemical conditions of the study site before the mesocosms deployment and during the experiment itself are also described, and a general overview of the papers published in this special issue is presented. All papers from the special issue could then refer to this one to avoid repeating the detailed mesocosms strategy (which was quite complex) in their paper**

**The present Synthesis paper aims at summarizing the major experimental and modelling results obtained during the project and described in the Special issue. We thus decided to divide this in 2 distinct papers**

Specific Comments p. 2, line 11: "a stable water mass" – Was turbulence measured?

**The turbulence has not been measured. We replaced the sentence by 'The sentence has been replaced by 'Triplicate large volume (~ 50 m$^3$) mesocosms were deployed in the tropical South West Pacific coastal ocean (New Caledonia) to isolate a water-mass with minimizing disturbance of ambient light and temperature conditions' in the revised version of the manuscript.….**

p. 3, line 5: ammonia is NH3, ammonium is NH4+

**"Ammonia" has been replaced by "ammonium" in the new version of the manuscript.**

p. 3, line 6: crops, not cultures?

**"Cultures" has been replaced by "crops" in the new version of the manuscript.**

p. 5, line 21: quantified, not qualified?

**"Qualified" has been replaced by "quantified" in the new version of the manuscript.**

p. 6, line 22: Eastern Tropical Pacific?

**We change to "Eastern Tropical North Pacific" as mentioned in White et al. (2012) in the new version of the manuscript.**

p. 8, line 17: 40 nM NO3- seems high to me. So does 0.1-0.15 ïA¸ g Chl a l-1

**The sentence has been replaced by 'The New Caledonian lagoon was chosen as it is a well-studied environment (Special issue Marine Pollution Bulletin 2010 (Grenz and LeBorgne,**

**2010)) submitted to high oceanic influence (Ouillon et al., 2010) and exhibiting typical**
**LNLC conditions during the summer season ($NO_3^-$ concentrations <0.04 µmol $L^{-1}$ and**
**chlorophyll a (Chl *a*) ~0.10-0.15 µg $L^{-1}$ (Fichez et al., 2010)'.**
Fig. 3: Why not plot particulate P and DOP?
**We chose to present in this figure mainly the plots related to the N dynamics as this is**
**what is specifically discussed in the manuscript. Particulate P and DOP are both presented**
**in the companion paper Berthelot et al. (2015) within the special issue.**
Fig. 3: units on (h) PON export seem to be incorrect
**Indeed, the units for PON export were wrong (should be µmol $d^{-1}$ instead of µmol $L^{-1}$). The**
**correction has been applied to the figure and its caption in the new version of the**
**manuscript.**
References cited:
Duhamel, S., Zeman, F., and Moutin, T.: A dual labelling method for the simultaneous
measurement of dissolved inorganic carbon and phosphate uptake by marine planktonic
species, Limnol. Oceanogr.-Meth., 4, 416–425, doi:10.4319/lom.2006.4.416, 2006.

[revised manuscript text omitted]

6[th] (austral summer). The New Caledonian lagoon was chosen as it is a well-studied environment (Special issue Marine Pollution Bulletin 2010 (Grenz and LeBorgne, 2010))

submitted to high oceanic influence (Ouillon et al., 2010) and  exhibiting typical

LNLC conditions during the summer season ($NO_3^-$ concentrations <0.04 µmol L$^-$

$^1$ and chlorophyll a (Chl *a*) ~0.10-0.15 µg L$^{-1}$ (Fichez et al., 2010). Primary productivity is N- limited throughout the year (Torréton et al., 2010), giving diazotrophs a competitive advantage. New Caledonian waters support high $N_2$ fixation rates (151-703 µmol N m$^{-2}$ d$^{-1}$, (Garcia et al., 2007)), as well as high *Trichodesmium* spp. (Dupouy et al., 2000; Rodier and

Le Borgne, 2010, 2008), and UCYN abundances (Biegala and Raimbault, 2008), therefore representing an ideal location to implement the VAHINE project and study the fate of DDN in the marine ecosystem.

DIP availability can control $N_2$ fixation in the southwestern Pacific (Moutin et al., 2008;

Moutin et al., 2005), hence the mesocosms were intentionally fertilized with ~0.8 µM DIP

($KH_2PO_4$) on the evening of day 4 to alleviate any potential DIP limitation and promote $N_2$

fixation and even diazotroph blooms for the purpose of the project.

The mesocosms used for this study are well suited for conducting replicated process studies on the first levels of the pelagic food web (Bonnet et al., 2016b; Guieu et al., 2010; Guieu et al., 2014). They are equipped with sediment traps allowing the collection of sinking material.

Due to the height of the mesocosms (15 m), they do not represent processes occurring in the full photic layer but allow studying the dynamics of C, N, P pools/fluxes and export associated with the plankton diversity in the same water mass, and comparing these dynamics.

before/after the DIP fertilization, and under contrasted conditions regarding the diazotroph community composition (cf below). Detailed surveys performed in LNLC environments revealed that temperature and light conditions are not affected by the presence of the mesocosms compared to surrounding waters (Bonnet et al., 2016b; Guieu et al., 2010; Guieu et al., 2014). These studies also revealed a good replicability and low variability  between stocks, fluxes and plankton diversity measurements among the replicate mesocosms. Hence, the discussion below will consider the average between the three mesocosms deployed in this study.

**2.2 Sampling strategy and logistics**

A complete description of the mesocosms design and deployment strategy is given in the introductory article (Bonnet et al., 2016b). In total, over 47 stocks, fluxes, enzymatic activities


[revised manuscript text omitted]

Diazotrophs transfer DDN to phytoplankton and heterotrophic prokaryotes via the dissolved N pool (DON and $NH_4^+$). During the maximal abundance of UCYN-C, these were responsible for 90±29 % of total $N_2$ fixation rates in the mesocosms (Bonnet et al., 2016a). During this period, the DDN released to the dissolved pool accounted for 7.1±1.2 to 20.6±8.1 % of gross $N_2$ fixation (Bonnet et al., 2016a) (based on the direct measurement of the isotopic signature ($^{15}N$) of the total dissolved N according to the denitrifying method (Knapp et al., 2005)) . This proportion is higher than that reported for UCYN-C in monospecific cultures using an equivalent method (1.0±0.3 to 1.3±0.2 % of gross $N_2$ fixation (Benavides et al., 2013a; Berthelot et al., 2015a).  At the same time as

UCYN-C bloomed, a diverse diazotroph community present in the mesocosms (Turk-Kubo
et al., 2015) also contributed to the  DDN release.
. Additionally, exogenous factors such as viral lysis (Fuhrman, 1999)
and sloppy feeding (O'Neil and Roman, 1992) occur in natural populations and could enhance
N release compared to the mono-culture studies. Here, we  demonstrate that natural
UCYN blooms may result in substantial DDN release to the marine environment.

The physiological state of cells probably plays a critical role in the quantity and availability of
DDN to the microbial communities as demonstrated in a study (applying identical
methodology) from two naturally-occurring blooms of *Trichodesmium* spp. in the same area
(New Caledonian lagoon) (Bonnet et al., Accepted). DDN release from these blooms was
slightly higher (bloom 1: $20\pm5$ to $48\pm5$ % and bloom 2: $13\pm2$ to $28\pm6$ % of gross $N_2$ fixation)
compared to UCYN-C (Bonnet et al., Accepted). A decaying *Trichodesmium* spp. bloom
(Bloom 1lead to high DDN release rates and high $NH_4^+$ accumulation
(up to 3.4 μM) in the dissolved pool. while we did not observe this in exponentially growing
*Trichodesmium* (Bloom 2).
. The importance of physiological status rather than
specific diazotroph types was further substantiated in earlier *Trichodesmium* culture
studies (Mulholland et al., 2004; Mulholland and Capone, 2000) and
similar DDN release between *Trichodesmium* spp. and three strains of UCYN-B
and C were found by Berthelot et al. (2015a).

Previous comparisons between gross and net $N_2$ fixation rates indicated high DDN release
rates for oceanic populations of *Trichodesmium* spp. (40-50 % of gross $N_2$ fixation on
average, and up to 97 %, (Mulholland, 2007) and references therein). The physiological status
of these populations may have influenced the fluxes. Furthermore, the values could reflect a
methodological overestimation due to the use of the $^{15}N_2$ bubble method (Großkopf et al.,
2012; Montoya et al., 1996) that may lead to greater differences between gross and net $N_2$
fixation (see introduction). Currently, direct measurement of the $^{15}N$ signature of the
dissolved N pool itself (either the TDN pool through the Knapp et al. (2005) method or both
the $NH_4^+$ and the DON using the Slawyk and Raimbault (1995) method) appears the preferred
method to accurately quantify the amount of DDN released by diazotrophs in the dissolved
pool (Berthelot et al., 2015a).

Once released in the form of $NH_4^+$ and/or DON, DDN can be taken up by surrounding planktonic communities. Experimental evidence from nanoSIMS experiments during VAHINE indicate that $21\pm4$ % of the $^{15}N_2$ fixed during the UCYN-C bloom was transferred to the non-diazotrophic plankton after 24 h of incubation (Bonnet et al., 2016a). Among these $21\pm4$ %, $18\pm3$ % was transferred to picoplankton (including both pico-phytoplankton and heterotrophic prokaryotes) and 3 % to diatoms (Fig. 5b), suggesting that picoplankton would be more competitive than diatoms using DDN, which is consistent with the increase in *Synechococcus* and pico-eukaryote abundances by a factor of two following the UCYN-C bloom (Leblanc et al., In review, 2016; Pfreundt et al., 2016). The short-term nanoSIMS experiment was performed on day 17, when pico- and nanoplankton dominated the phytoplankonic biomass and diatom abundances declined probably due to DIP limitation (Leblanc et al., In review, 2016). Picoplankton can efficiently utilize low DIP concentrations (Moutin et al., 2002) and/or can use alternative DOP sources (Benitez-Nelson and Buesseler, 1999) . This,  may explain why  picoplankton were the first beneficiaries of the DDN from UCYN-C  specifically from days 17-23, although we cannot exclude that diatoms had also benefited from the DDN from UCYN-C  earlier in the experiment (between days 10-11 and days 15-16 when they reached bloom values of ~100 000 cells $L^{-1}$).

A significant increase of both PP and BP during P2 (Fig. 2) suggests that both autotrophic and heterotrophic communities benefited from the DDN (Bonnet et al., 2016a). Calculations based on C:N molar ratios show that $N_2$ fixation may have provided ~30 % of the N demand of the N-limited bacteria during P2 (compared to ~20 % during P1), the rest  provided by detritus and DON (Van Wambeke et al., Accepted), which concentrations decreased during the 23 days (Berthelot et al., 2015b). Throughout VAHINE, the  biological system inside the mesocosms was net autotrophic  with an upper error limit close to the metabolic balance between autotrophy and heterotrophy (Van Wambeke et al., Accepted). The  relationships between BP and $N_2$ fixation rates were weak (during P2) or absent (during P1) yet tightly coupled  between BP and Chl *a* concentrations, and between BP and PP. This suggests that $N_2$ fixation stimulated autotrophic communities and these subsequently  fueled heterotrophic prokaryotes through the production and release of dissolved organic matter including C (DOC) (Van Wambeke et al., Accepted).

In a recent study performed at the VAHINE study site, (Berthelot et al., In review, 2016) compared the DDN transfer efficiency to several groups of non-diazotrophic plankton as a function of the diazotroph groups dominating the community (*Trichodesmium* spp. *versus* UCYN-B *versus* UCYN-C). Simulated blooms of *Trichodesmium* spp., UCYN-B and UCYN-C grown in culture added to ambient lagoon communities reveal that the primary route of transfer of DDN towards non-diazotrophs is $NH_4^+$, and DON mainly accumulates in the dissolved pool, whatever the diazotroph considered. In all cases, the presence of diazotrophs stimulated biomass production of non-diazotrophs, with heterotrophic prokaryotes the main DDN beneficiaries  followed by diatoms and picophytoplankton. NanoSIMS analyses revealed that heterotrophic prokaryotes were highly [15]N-enriched, confirming they can directly benefit from the DDN (Berthelot et al., In review, 2016). Further studies are needed to study the indirect stimulation of heterotrophic prokaryotes through the release of DOC by diazotrophs and non-diazotrophic phytoplankton that were -stimulated by the DDN.

Similar experiments ([15]$N_2$ labelling, flow cytometry cell sorting and nanoSIMS) performed on three naturally-occurring *Trichodesmium* spp. blooms in the southwestern Pacific illustrated that DDN was predominantly transferred to diatoms  (Bonnet et al., Accepted). These results  indicate that  the extensive oceanic blooms of *Trichodesmium* spp.  can contribute to a large indirect downward flux of organic matter by promoting large cells (e.g., diatoms and dinoflagellates) characterized by efficient export rates (Nelson et al., 1995, Bonnet et al., Accepted; Devassy et al., 1979; Lenes et al., 2001).

Direct export flux of *Trichodesmium* spp. blooms may also occur in cases where rapid (< 2 d) bloom mortality occurs via a programmed cell death (PCD)  (Berman-Frank et al., 2004; Berman-Frank et al., 2007). PCD in *Trichodesmium* spp. is  characterized by the loss of buoyancy (collapse of gas vesicles) and increased production of TEP and aggregation leading to enhanced and massive vertical flux (Bar-Zeev et al., 2013). A *Trichodesmium* spp. bloom that occurred outside the VAHINE mesocosms on days 23-24 displayed mechanistic features of PCD including mass mortality within 24 h, loss of gas vesicles, and high production of TEP (Spungin et al., In review, 2016). While we could not directly quantify the export flux as no sediment traps were deployed in the lagoon water outside the mesocosms, the characteristics of the bloom, lack of grazer influence and the demise of biomass suggests this would lead to high rates of export (Spungin et al., In review,

2016) as demonstrated in culture simulations (Bar-Zeev et al., 2013) (Fig 5c).

**4.2.2 DDN transfer to zooplankton**

DDN transfer to zooplankton may either be direct through the ingestion of diazotrophs, or indirect, i.e. mediated through the release of dissolved DDN by diazotrophs taken up by heterotrophic and autotrophic plankton and subsequently grazed by zooplankton. During the

VAHINE experiment, the percent contribution of DDN to zooplankton biomass averaged 30

% (range = 15 to 70 %) (Hunt et al., Accepted), which is in upper range of values reported from high $N_2$ fixation areas such as the subtropical north Atlantic (Landrum et al., 2011;

Mompean et al., 2013; Montoya et al., 2002a), the Baltic Sea (Sommer et al., 2006; Wannicke et al., 2013b), and the pelagic waters off the New Caledonian shelf (Hunt et al., 2015).

During VAHINE all four of the qPCR targeted diazotrophs (*Trichodesmium* spp., het-1, het-2,

UCYN-C) were found in zooplankton guts indicating a direct grazing of these four phylotypes (Hunt et al., Accepted). Overall, the most frequently detected targets were het-1 (during P1;

17 to 180 *nifH* copies copepod$^{-1}$) and UCYN-C (during P2; 7 to 50 *nifH* copies copepod$^{-1}$), i.e.

the most abundant phylotypes encountered in the mesocosms during P1 and P2, respectively.

However, *Trichodesmium* spp. and het-2 were also detected at relatively high abundances in copepod guts (~280 *nifH* copies copepod$^{-1}$) despite their low abundance in the mesocosms, suggesting selective feeding and a possible top down control through zooplankton grazing for these two phylotypes.

Direct and efficient zooplankton grazing on UCYN-C was further substantiated by targeted grazing experiments during VAHINE which consisted of $^{15}N_2$-labeled bottle incubations of freshly collected zooplankton in the presence of natural phytoplankton assemblages. The $^{15}N_2$

label was taken up by the diazotroph in the incubation bottles and used as a marker of zooplankton diazotroph ingestion and/or ingestion of non-diazotrophic plankton grown on

DDN. Zooplankton were highly $^{15}N$ enriched after 72 h of incubation during the UCYN-C

bloom (P2), slightly enriched during P1 when DDAs dominated to diazotrophic community, and not enriched at all when a *Trichodesmium* spp. bloom was encountered outside the mesocosms during P2 (Hunt et al., Accepted). This was a surprising finding given that het-1, and to a lesser extent *Trichodesmium* spp. were detected in copepod guts, and would suggest that UCYN-C are much more efficiently transferred to zooplankton compared to DDAs and


[revised manuscript text omitted]

It is likely that during the experiment, TEP release favored aggregation and accumulation of particles and subsequently enhanced vertical flux from the different compartments in the water column. To represent the latter phenomenon, we considered in the model that 10 % of the living and non-living compartments were allowed to sink after day 10 (see Gimenez et al. (2016) for more details). Since this extra aggregation is mainly attributable to diazotrophs, it was not represented in the NOFIX simulation. However, we ran a third simulation (not shown) to further analyze the excess of C export in the REF simulation as compared to the NOFIX one (Fig. 8). This third simulation is intermediate between the REF and the NOFIX simulations in that sense that only the $N_2$ fixation capability by diazotrophs is removed (but aggregation processes are still represented). This simulation indicated that C export is nearly equal to that of the REF simulation after 25 days (they differ by only 2.9 %), thereby suggesting that during the 25 first days, the suppression of $N_2$ fixation does not significantly impact carbon export fluxes. This further suggests that the higher C export in the REF simulation during P2 (Fig.8) is mainly due to aggregation processes mediated by diazotrophs-derived TEP release and the subsequent export of diazotrophs (Berman-Frank et al., 2016; Bonnet et al., 2015a). However, beyond day 25, the difference in C export between the REF

[revised manuscript text omitted]

may develop extensively under high temperature and $p$CO$_2$ conditions (Dutkiewicz et al.,

2015), while others such as UCYN-A would not be affected (Law et al., 2012). The results from the VAHINE project revealed that the diazotroph community composition can impact  the  planktonic food web structure and composition in the surface ocean, and  also affects the efficiency of particulate matter export to depth. Thus, current and predicted global changes require further knowledge and understanding of the fate and implications of changing scenarios of $N_2$ fixation in the future oceans.

**Acknowledgements**

Funding for this research was provided by the Agence Nationale de la Recherche (ANR
starting grant VAHINE ANR-13-JS06-0002), the INSU-LEFE-CYBER program, GOPS and
IRD. The authors thank the captain and crew of the R/V *Alis*. We acknowledge the SEOH
diver service from Noumea, as well as the technical service of the IRD research center of
Noumea for their helpful technical support together with C. Guieu, J.-M. Grisoni and F. Louis

for the mesocosm design and the useful advice. Partial funding to IBF was provided through a collaborative grant with SB from the Israel Ministry of Science and Technology (MOST) Israel and the High Council for Science and Technology (HCST)-France, and a German-Israeli Foundation for Scientific Research and Development (GIF) GIF grantgrant No. 1133-13.8/2011.

**Figure legends.**

**Figure 1.** Study site of the VAHINE experiment. Location map of New Caledonia in the Southwestern Pacific (a), Map of the Noumea lagoon showing the location of mesocosms at the entrance of the lagoon, 28 km off the coast (b).

**Figure 2.** View of the mesocosms from above (a), from the seafloor (b) and view of the sediment traps that collect sinking particles (c) (Photos credits: J.M. Boré and E. Folcher, IRD).

**Figure 3.** Evolution of sea surface temperature (°C) (a), $NO_3^-$ ($\mu$mol $L^{-1}$) (b), DIP ($\mu$mol $L^{-1}$) (c), Chl a ($\mu$g $L^{-1}$) (d), $N_2$ fixation rates (nmol N $L^{-1}$ $d^{-1}$) (e), PON concentrations ($\mu$mol $L^{-1}$) (f), DON concentrations ($\mu$mol $L^{-1}$) (g) and PON export ($\mu$mol $d^{-1}$) (h) over the 23 days of the VAHINE mesocosm experiment. Lines represent the average of the three mesocoms and shaded areas represent the measured min and max values.

**Figure 4.** Upper panel: Diazotroph community composition in the VAHINE mesocosm experiment during the experimental period. *nifH*-based abundances were summed for each sampling day to determine the percent contribution to the total diazotroph community from each major phylotype (data from Turk-Kubo et al. (2015)). Bottom panel: simplified evolution of the major standing stocks, rates and plankton abundances measured during P1 (days 5 to 14) and P2 (days 15 to 23). Protocols for each parameter measurements are  described in Berthelot et al. (2015), Bonnet et al. (2016a,b), Van Wambeke et al., (2016), Berman-Frank et al., (2016),  Leblanc et al. (2016), Turk-Kubo et al., (2015) and Hunt et al., (2016). Squares are represented in green when a significant (p<0.05) increase was observed between each period (i.e. between P0 and P1 or between P1 and P2, Kruskall-Wallis test, $\alpha$=0.05), in red when a significant (p<0.05) decrease was observed and in grey when no significant change was observed between the different periods.

**Figure 5.** Summary of the simplified pathways of the potential DDN transfer in the first trophic level of the food web and potential of direct *versus* indirect export of particulate matter for DDAs (a), UCYN-C (b) and *Trichodesmium* (c). DDN transfer data from (Bonnet et al., Accepted; Bonnet et al., 2016a)

**Figure 6.** Evolution of PP ($\mu$mol C $L^{-1}$ $d^{-1}$) (a) and bacterial production (ng C $L^{-1}$ $h^{-1}$) in the REF simulation ( blue line) and the NOFIX simulation (black line) (i.e. when the $N_2$ fixation process is removed).

**Figure 7.** Evolution of plankton abundances (cells $L^{-1}$) in the REF simulation (blue line) and the NOFIX simulation (black line) (i.e. when the $N_2$ fixation process is removed). TRI: *Trichodesmium* spp., UCYN: UCYN-C, BAC: heterotrophic bacteria, PHYS: small phytoplankton, HNF: heterotrophic nanoflagellates.

**Figure 8.** Evolution of C content collected in the mesocosm particle traps (mmol C) in the REF simulation (blue line) and the NOFIX simulation (black line) (i.e. when the $N_2$ fixation process is removed).

**References cited**

[revised manuscript text omitted]

1606 Mulholland, M. R. and Capone, D. G.: The nitrogen physiology of the marine N-2-fixing cyanobacteria
1607 Trichodesmium spp., Trends in Plant Science, 5, 148-153, 2000.
1608 O'Neil, J. M.: Grazer interactions with nitrogen-fixing marine Cyanobacteria: adaptation for N-
1609 acquisition?, Bull. Inst. Oceanogr. Monaco, 19, 293-317, 1999.
1610 O'Neil, J. M., Metzler, P., and Glibert, P. M.: Ingestion of $^{15}N_2$-labelled *Trichodesmium*, and
1611 ammonium regeneration by the pelagic harpacticoid copepod *Macrosetella gracilis*, Marine Biology,
1612 125, 89-96, 1996.
1613 O'Neil, J. and Roman, M. R.: Grazers and Associated Organisms of *Trichodesmium*. In: Marine Pelagic
1614 Cyanobacteria: *Trichodesmium* and other Diazotrophs, Carpenter, E. J., Capone, D.G., and Rueter, J.G.
1615 (Ed.), NATO ASI Series, Springer Netherlands, 1992.
1616 Ouillon, S., Douillet, P., Lefebvre, J. P., Le Gendre, R., Jouon, A., Bonneton, P., Fernandez, J. M.,
1617 Chevillon, C., Magand, O., Lefèvre, J., Le Hir, P., Laganier, R., Dumas, F., Marchesiello, P., Bel Madani,
1618 A., Andréfouët, S., Panché, J. Y., and Fichez, R.: Circulation and suspended sediment transport in a
1619 coral reef lagoon: The south-west lagoon of New Caledonia, Marine Pollution Bulletin, 61, 269-276,
1620 2010.
1621 Pfreundt, U., Spungin, D., Berman-Frank, I., Bonnet, S., and Hess, W. R.: Global analysis of gene
1622 expression dynamics within the marine microbial community during the VAHINE mesocosm
1623 experiment in the South West Pacific, Biogeosciences Discussions, doi: doi:10.5194/bg-2015-564, In
1624 review, 2016. In review, 2016.
1625 Pfreundt, U., Van Wambeke, F., Caffin, M., Bonnet, S., and Hess, W. R.: Succession within the
1626 prokaryotic communities during the VAHINE mesocosms experiment in the New Caledonia lagoon,
1627 Biogeosciences, 13, 2319-2337, 2016.
1628 Rodier, M. and Le Borgne, R.: Population and trophic dynamics of Trichodesmium thiebautii in the SE
1629 lagoon of New Caledonia. Comparison with T. erythraeum in the SW lagoon, Marine Pollution
1630 Bulletin, 61, 349-359, 2010.
1631 Rodier, M. and Le Borgne, R.: Population dynamics and environmental conditions affecting
1632 Trichodesmium spp. (filamentous cyanobacteria) blooms in the south-west lagoon of New Caledonia,
1633 Journal of Experimental Marine Biology and Ecology, 358, 20-32, 2008.
1634 Scharek, R., Latasa, M., Karl, D. M., and Bidigare, R. R.: Temporal variations in diatom abundance and
1635 downward vertical fux in the oligotrophic North Pacific gyre, Deep Sea Research Part I, 46, 1051-
1636 1075, 1999a.
1637 Sharek, R. M., Tupas, L. M., and Karl, D. M.: Diatom fluxes to the deep sea in the oligotrophic North
1638 Pacific gyre at Station ALOHA, Marine and Ecological Progress Series, 82, 55-67, 1999b.
1639 Sipler, R. A., Bronk, D. A., Seitzinger, S. P., Lauck, R. J., McGuinness, L. R., Kirkpatrick, G. J., Heil, C. A.,
1640 Kerkhof, L. J., and Schofield, O. M.: *Trichodesmium*-derived dissolved organic matter is a source of
1641 nitrogen capable of supporting the growth of toxic red tide *Karenia brevis*, Marine and Ecological
1642 Progress Series, 483, 31-45, 2013.
1643 Slawyk, G. and Raimbault, P.: Simple procedure for simultaneous recovery of dissolved inorganic and
1644 organic nitrogen in $^{15}N$-tracer experiments and improving the isotopic mass balance, Marine and
1645 Ecological Progress Series, doi: doi:10.3354/meps124289, 1995. 1995.
1646 Sommer, S., Hansen, T., and Sommer, U.: Transfer of diazotrophic nitrogen to mesozooplankton in
1647 Kiel Fjord, Western Baltic Sea: a mesocosm study, Marine Ecology Progress Series, 324, 105-112,
1648 2006.
1649 Spungin, D., Pfreundt, U., Berthelot, H., Bonnet, S., AlRoumi, D., Natale, F., Hess, H. R., Bidle, K. D.,
1650 and Berman-Frank, I.: Mechanisms of Trichodesmium bloom demise within the New Caledonia

Lagoon during the VAHINE mesocosm experiment, doi: doi:10.5194/bg-2015-613, In review, 2016. In review, 2016.

Torréton, J.-P., Rochelle-Newall, E., Pringault, O., Jacquet, S., Faure, V., and Briand, E.: Variability of primary and bacterial production in a coral reef lagoon (New Caledonia), Marine Pollution Bulletin, 61, 335, 2010.

Turk-Kubo, K. A., Frank, I. E., Hogan, M. E., Desnues, A., Bonnet, S., and Zehr, J. P.: Diazotroph community succession during the VAHINE mesocosms experiment (New Caledonia Lagoon), Biogeosciences, 12, 7435-7452, 2015.

Van Wambeke, F., Pfreundt, U., Barani, A., Berthelot, H., Moutin, T., Rodier, M., Hess, W., and Bonnet, S.: Heterotrophic bacterial production and metabolic balance during the VAHINE mesocosm experiment in the New Caledonia lagoon Biogeosciences, 12, 19861-19900, Accepted.

Walsby, A. E.: The gas vesicles and buoyancy of *Trichodesmium*, Marine Pelagic Cyanobacteria: Trichodesmium and other Diazotrophs, 1992. 141-161, 1992.

Wannicke, N., Korth, F., Liskow, I., and Voss, M.: Incorporation of diazotrophic fixed $N_2$ by mesozooplankton - Case studies in the southern Baltic Sea, Journal of Marine Systems, 117-118, 1-13, 2013a.

Wannicke, N., Korth, F., Liskow, I., and Voss, M.: Incorporation of diazotrophic fixed $N_2$ by mesozooplankton - Case studies in the southern Baltic Sea, Journal of Marine Systems, 117-118, 1-13, 2013b.

White, A. E., Foster, R. A., Benitez-Nelson, C. R., Masqué, P., Verdeny, E., Popp, B. N., Arthur, K. E., and Prahl, F. G.: Nitrogen fixation in the Gulf of California and the Eastern Tropical North Pacific, Progess in Oceanography, 109, 1-17, 2012.